# WildLong: Synthesizing Realistic Long-Context Instruction Data at Scale

## Abstract

Large language models (LLMs) with extended context windows enable tasks requiring extensive information integration but are limited by the scarcity of high-quality, diverse datasets for long-context instruction tuning. Existing data synthesis methods focus narrowly on objectives like fact retrieval and summarization, restricting their generalizability to complex real-world tasks. We introduce WildLong, a framework for generating diverse, scalable, and realistic instruction-response datasets tailored to long-context tasks. WildLong extracts meta-information from real user queries, models co-occurrence relationships via graph-based methods, and employs adaptive generation to produce scalable data. It extends beyond single-document tasks to support multi-document reasoning, such as cross-document comparison and aggregation. Our models, finetuned on 150K instruction-response pairs synthesized using WildLong, surpasses existing open-source long-context-optimized models across benchmarks while maintaining strong performance on short-context tasks without incorporating supplementary short-context data. By generating a more diverse and realistic long-context instruction dataset, WildLong enhances LLMs' ability to generalize to complex, real-world reasoning over long contexts, establishing a new paradigm for long-context instruction tuning.

## 1 Introduction

The growing demand for AI systems capable of processing and reasoning over extensive information has driven the development of large language models (LLMs) with significantly expanded context windows (Dubey et al., 2024; Achiam et al., 2023; Team et al., 2024). Among long-context tasks, needle-in-a-haystack (NIAH) (Kamradt, 2023) retrieval—where models locate specific information within long contexts—has emerged as a relatively simple benchmark, with previous work showing that continued pretraining on long-context data significantly improves NIAH performance (Fu et al., 2024; Hsieh et al., 2024; Li et al., 2024c). However, while many LLMs excel at NIAH, they struggle with more complex tasks, such as passage ranking and dialogue analysis, which require reasoning and synthesis across extended contexts (Hsieh et al., 2024; Yen et al., 2025; Zhang et al., 2024b; Levy et al., 2024; Vodrahalli et al., 2024; Li et al., 2024b). The ability to reason over long contexts is essential for real-world applications, such as legal document analysis and book review (Liu et al., 2024b; Karpinska et al., 2024; Xu et al., 2024b;c; Jimenez et al., 2024; Wang et al., 2024a).

A major bottleneck in enhancing long-context reasoning is the lack of high-quality instruction tuning data. Unlike short-context tuning, which benefits from abundant human-annotated data, manually constructing long-context instruction data is impractical due to the complexity of reasoning over extended contexts. Existing methods rely on data synthesis using LLMs (Dubey et al., 2024; An et al., 2024b; Bai et al., 2024; Xiong et al., 2024; 2025b). For instance, prior approaches (Xiong et al., 2024; Bai et al., 2024) generate long-context instruction-tuning data by extracting short text spans from long documents, synthesizing question-answer pairs based on these snippets, and incorporating the full document during training. Other approaches, such as Llama-3.1 (Dubey et al., 2024), further utilize hierarchical summarization to construct long-context datasets. While effective at leveraging models' short-context capabilities for data generation, these methods primarily focus on fact extraction and summarization. This narrow scope limits the diversity and generalizability of the resulting data, leaving critical gaps in supporting more complex and realistic tasks.

To address this, we propose WildLong, a scalable framework for generating diverse and realistic instruction-response pairs for long-context reasoning. Our approach integrates *meta-information extraction*, *graph-based modeling*, and *adaptive instruction-response generation*. The pipeline of our framework is illustrated in Figure 1. First, we extract meta-information (e.g., user intent, task type, constraints) from real-world user-chatbot conversations to ground instruction-response pairs in realistic scenarios. To enhance diversity and scalability, we model the extracted meta-information as a graph, where nodes represent individual meta-information values and edges capture their co-occurrence frequencies. Random walks on this graph generate novel combinations of meta-information, introducing diverse and varied instruction templates. Finally, we pair these templates with long-context examples from the pretraining corpus in adaptive instruction-response generation stage, ensuring large-scale, diverse dataset creation.

We fine-tune Mistral-7B-Instruct-v0.2 and Llama-3.1-8B-Instruct on 150K synthesized instruction-response pairs and evaluate them on various long-context benchmarks with input lengths up to 128K tokens. Notably, our fine-tuned Mistral-7B model achieves a substantial $+14.7$ improvement on the RULER benchmark (Hsieh et al., 2024), while our Llama-3.1-8B model performed competitively with much larger models, scoring $84.1$ on RULER (vs. $85.1$ for Llama-3.1 70B) and $6.8$ on LongBench-Chat (Bai et al., 2024) (vs. $6.7$ for Llama-3.1 70B). Importantly, our fine-tuned models retain short-context performance without fine-tuning on additional short-context data, further demonstrating the robustness and generalizability of our synthetic data.

## 2 PROPOSED METHOD

### 2.1 META INFORMATION EXTRACTION

We leverage the WildChat dataset (Zhao et al., 2024b), a large corpus of user-chatbot conversations, and focus specifically on single-turn conversations involving long contexts. From each filtered long conversation, we extract 13 fields of meta-information[1] that capture key attributes relevant to long-context instructions. These fields encompass essential aspects of the interaction, ensuring a comprehensive representation of user intent, contextual nuances, and task-specific requirements. We prompt `GPT-4` to extract meta information from each conversation. For example, task like "extract details" for informational articles is explicitly labeled. Context like "preparing for a presentation" is extracted for professional text. The extracted meta-information serves as a structured foundation for subsequent stages of our methodology.

### 2.2 GRAPH CONSTRUCTION

Instructions are generally document-type-specific, necessitating the construction of separate graphs for each document type. Document types are initially extracted as free-form values during meta-information extraction. We then apply K-Means clustering to identify ten document types, balancing specificity and generalization[2]. For each document type $d$, we construct an undirected graph $G_d = (\mathbb{V}_d, E_d)$ to model the co-occurrence among meta information values extracted from user-chatbot conversations. This graph represents the interactions between meta information fields and facilitates the systematic exploration of realistic and diverse combinations for instruction generation. The construction process is detailed as follows.

**Nodes** Each node corresponds to a unique value of a meta information field. Let $\mathbb{M} = \{m_1, m_2, \ldots, m_{11}\}$ denote the set of 11 meta information fields used to construct the graph[3]. The set of nodes $\mathbb{V}_d$ is defined as:

$$\mathbb{V}_d = \big\{ v \mid v \text{ is a value of some field } m_i \in \mathbb{M} \text{ in any conversation for document type } d \big\},$$

which collectively capture all unique meta information values observed for a specific document type.

---

[1]The meta-information includes: document type, tasks or requests, user intention, user profile, language style, context, knowledge/commonsense involved for user, knowledge/commonsense involved for chatbot, long context capability involved, output format, sentiment, constraint of the request, simplified instruction.

[2]Figure 6 shows the distribution of these types.

[3]The "document type" field is used to classify documents such that we can construct a separate graph for each document type. The "simplified instruction" field is used as a demonstration when generating instructions based on paths, see Section 2.4.

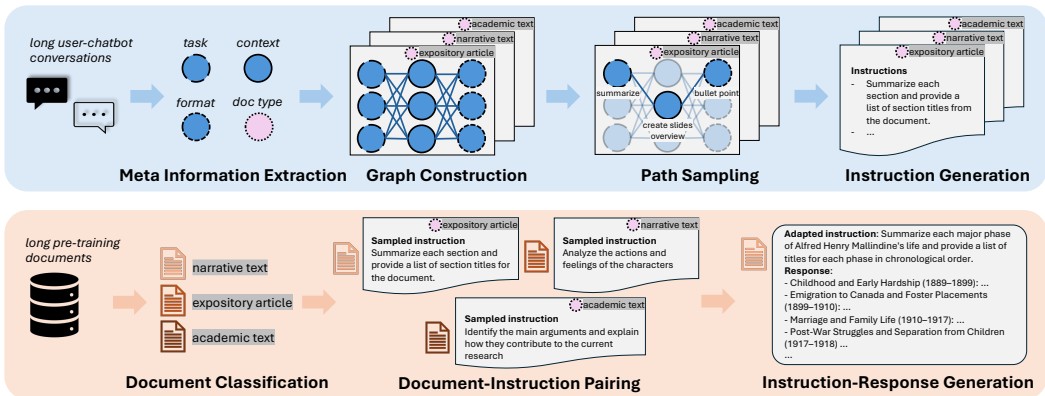

Figure 1: Overview of the two-stage WildLong Framework. Stage 1 extracts meta-information from real-world user-chatbot conversations, classifies documents by type, constructs graphs to represent meta-information relationships, and samples paths to generate tailored instructions. Stage 2 pairs long documents with these instructions, generating instruction-response pairs by rewriting the instructions and answering based on the document context.

**Edges** Edges connect meta-information values that co-occur in the same conversation and belong to different fields. Formally, an edge $(v, u) \in E_d$ exists if:

- $v$ is a value of field $m_i \in \mathbb{M}$,
- $u$ is a value of field $m_j \in \mathbb{M}$, where $i \neq j$, and
- $v$ and $u$ co-occur in at least one conversation for document type $d$.

For each conversation, the extracted meta information values from the 11 fields are interconnected, forming a fully connected bipartite subgraph.

**Edge Weights** The weight of an edge $(v, u) \in E_d$ reflects the frequency of co-occurrence of $v$ and $u$ across all conversations for document type $d$. The edge weight is computed as: $w(v, u) = \log(f_{\text{co}}(v, u) + \varepsilon)$, where $f_{\text{co}}(v, u)$ is the raw count of co-occurrences, and $\varepsilon$ is a small constant for numerical stability. Logarithmic scaling reduces the impact of highly frequent pairs while preserving distinctions among lower-frequency edges. By capturing the variety and co-occurrence patterns of meta-information values, these document-type-specific graphs provide a foundation for generating realistic, meaningful, and diverse instruction paths.

## 2.3 META INFORMATION PATH SAMPLING

To guide instruction generation with realistic and diverse criteria, we sample structured combinations of meta information values. Since meta information fields interact in complex ways, manually enumerating all meaningful combinations is infeasible. Instead, we generate sampled paths $\hat{P} = \{v_1, v_2, \ldots, v_k\}$ that represent meta information combinations by employing a weighted random walk algorithm on $G_d$. The walk starts with an initial node $v_1 \in \mathbb{V}_d$ chosen from a uniformly sampled meta information category $m_c \in \mathbb{M}$. At each step $t$, the walk transitions from the current node $v_t$ to a neighboring node $v_{t+1}$, which belongs to a different, unvisited meta information category. The transition probability from $v_t$ to $v_{t+1}$ is determined by edge weights:

$$p(v_{t+1} \mid v_t) = \frac{\exp(w(v_t, v_{t+1}))}{\sum_{v_k \in \mathcal{N}(v_t)} \exp(w(v_t, v_k))}, \tag{1}$$

where $w(v_t, v_{t+1})$ is the edge weight, and $\mathcal{N}(v_t)$ is the set of neighbors of $v_t$. The walk continues for up to $N$ steps, producing a path spanning $N$ distinct meta information fields. Based on our preliminary experiments on instruction synthesis, we determined that $N = 6$ strikes the right balance, where larger values of $N$ introduce overly restrictive criteria, making instruction generation challenging and prone to producing convoluted instructions joined by "and", while smaller values of $N$ result in overly simple instructions with limited complexity. We conducted an ablation study to validate the effect of $N$ in Appendix C.2. By leveraging edge weights to guide transitions, the algorithm captures realistic co-occurrence patterns, enabling the scalable synthesis of diverse instruction templates,

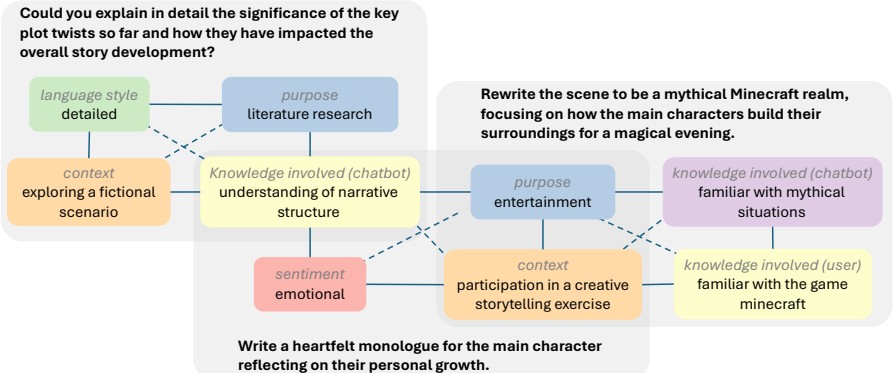

Figure 2: Examples of instructions generated from sampled paths in a narrative text graph. Solid lines represent connections within paths, while dotted lines show node interconnections during graph construction. A random walk algorithm produces diverse instructions by combining nodes. For instance, the knowledge node "understanding of narrative structure" and the context node "participation in a creative storytelling exercise" appear in multiple paths but result in distinct instructions due to varying other meta information.

while maintaining flexibility to explore less frequent connections. We empirically validated this sampling strategy, showing it significantly outperforms both greedy and uniform sampling baselines in Appendix C.1 These paths serve as structured templates to generate diverse and representative instructions for long-context tasks.

## 2.4 INSTRUCTION GENERATION WITH PATHS

To synthesize instructions aligned with the sampled meta-information paths, we prompt GPT-4 with a one-shot demonstration. GPT-4 generates natural language instructions that follow the criteria defined by the meta information fields in the sampled path[4]. Figure 2 illustrates instructions generated from sampled paths in a narrative text graph. By combining different meta-information values through the random walk algorithm, diverse instructions are synthesized. To quantitatively validate that our synthesized instructions reflect real-world user needs, we compared their task distribution against the original conversations in WildChat in Appendix F. This analysis confirmed a strong alignment, demonstrating that our graph-based approach successfully preserves the nature of user queries.

## 2.5 INSTRUCTION-RESPONSE PAIR GENERATION

Once the instructions are generated, we pair them with long documents sampled from the SlimPajama[5] dataset (Soboleva et al., 2023). SlimPajama's wealth of long documents makes it well-suited for tasks requiring extensive context. As instructions are document-type-specific, we first classify sampled documents into one of ten predefined document types using a custom classifier[6].

To align document distributions with realistic user queries, we resample SlimPajama's long documents to match the document type distribution of WildChat's long conversations. Documents are then paired with instructions generated from the graph corresponding to their type. To make the instructions more contextually grounded, the sampled instruction and paired document are provided as input to GPT-4, which generates an adapted instruction tailored to the document, and a corresponding response. This ensures the final instruction-response pairs are coherent, relevant, and reflective of the document's context. To ensure high fidelity, we implemented a rigorous quality control protocol throughout the generation process. This included iterative prompt refinement and a manual audit of the final instruction-response pairs for factual correctness, hallucination, ambiguity and safety, as detailed in Appendix F. We present concrete examples generated with WildLong in Appendix H.

---

[4]Details about how to select the demonstration can be found in Appendix B.3.

[5]SlimPajama is an open-source reproduction of the LLaMA pretraining data mixture (Touvron et al., 2023).

[6]Details about the classifier are provided in Appendix B.2.

## 2.6 EXTENDING INSTRUCTIONS TO MULTI-DOCUMENT SETTINGS

We observe that the filtered WildChat dataset predominantly contains instructions for single-document contexts, with limited coverage of multi-document tasks. To address this gap, we extend our method to generate instructions suitable for multi-document settings by adapting the extracted meta information and graph-based framework. The extension begins with modifying the "tasks or requests" field in the meta information to reflect multi-document requirements while keeping other fields unchanged. Each single-document task node is rewritten to explicitly handle information across multiple documents using `GPT-4`. For instance, a task like "Summarize the key points of the document" is transformed into "Summarize and compare the key points across multiple documents."

We then construct document-type-specific graphs for multi-document tasks and use the same random walk algorithm to sample paths and generate instructions. The graph construction, path sampling, and instruction synthesis steps remain consistent with the single-document setting. During the document-instruction pairing stage, pairs of documents of the same type are sampled from the SlimPajama dataset, concatenated, and paired with a multi-document instruction of the same type. These concatenated documents, along with the paired instruction, are input to `GPT-4` to generate a refined, contextually aligned instruction and corresponding response. By incorporating these modifications, our method systematically generates instructions and responses that support multi-document reasoning tasks, expanding the diversity and applicability of the dataset. We study the effect of multi-document data in Appendix D. The prompts used in the WildLong framework are detailed in Appendix I.

## 3 EXPERIMENTS

We evaluate our framework comprehensively on long- and short-context benchmarks. This section outlines implementation details, compares our method with baseline and specialized long-context optimized models, benchmarks against existing long-context supervised fine-tuning (SFT) datasets, and presents ablation studies to analyze the contributions of essential components in our framework.

### 3.1 IMPLEMENTATION DETAILS

**Data Curation** We filter single-turn WildChat conversations exceeding 2K tokens, yielding 32K instances. We then filter long-context documents from the SlimPajama corpus into two subsets: single-document (2K–30K tokens) and multi-document (2K–20K tokens). For multi-document, we pair two same-type documents and concatenate them. We sample 100K single-document and 50K multi-document examples, totaling 150K samples.

**Training Details.** We fine-tune Mistral-7B-Instruct-v0.2 and Llama-3.1-8B-Instruct using our curated dataset. For Mistral-7B-Instruct-v0.2, we adjust the RoPE base from 1e6 to 1e7 to support longer positional embeddings[7]. Both models are optimized using the Adam optimizer, with learning rates of 1e-6 and 5e-7 respectively. Training is conducted for 2 epochs with a batch size of 512[8].

### 3.2 BASELINES

**Proprietary Long-Context Models.** We include two proprietary long-context models `Gemini-1.5-Pro` and `GPT-4` as upper bounds due to their strong long-context performance.

**Open-Sourced Pretrained Long-Context Models.** Additionally, we evaluate open-source pretrained language models with long-context capabilities, including GLM4-9B (GLM et al., 2024), Yi-34B (AI et al., 2024), Llama3.1-70B (Dubey et al., 2024), Phi-3-medium (Abdin et al., 2024), and Qwen2.5 (Yang et al., 2024).

**Specialized Long-Context Optimized Models.** We compare our approach to specialized long-context LLMs that extend or optimize model capabilities for long inputs. FILM (An et al., 2024b) and ChatQA-2 (Xu et al., 2025a) fine-tunes Mistral-7B-Instruct-v0.2 and Llama-3-8B with synthetic

---

[7]Increasing RoPE base enables model to support longer context. More details can be seen in Appendix G.

[8]More details about computational budget and and infrastructure can be found in Appendix B.5.

Table 1: Main evaluation results of our models on RULER, HELMET and Longbench-Chat compared with baselines. Results on RULER and HELMET are averaged over sequence lengths ranging from 4K to 128K and 8K to 128K respectively.

| Models | Size | RULER | | | | | HELMET | | | | | | | Longbench |
|---|---|---|---|---|---|---|---|---|---|---|---|---|---|---|
| | | NIAH | VT | Agg | QA | Avg | RAG | ICL | Cite | Rank | QA | Summ | Avg | Chat |
| *Proprietary Long-Context Models* | | | | | | | | | | | | | | |
| Gemini-1.5-Pro | - | 99.7 | 99.9 | 96.6 | 77.2 | 93.4 | 72.1 | 78.8 | 44.5 | 69.0 | 47.6 | 38.5 | 58.4 | 7.6 |
| GPT-4 | - | 95.4 | 99.9 | 93.4 | 70.3 | 89.8 | 70.6 | 65.1 | 24.9 | 53.4 | 47.7 | 32.6 | 49.1 | 8.4 |
| *Open-Sourced Pretrained Long-Context Models* | | | | | | | | | | | | | | |
| GLM-4-1M | 9B | 98.2 | 99.4 | 72.2 | 69.4 | 84.8 | 67.9 | 77.3 | 31.4 | 41.7 | 44.2 | 28.8 | 48.6 | 5.9 |
| Yi-200k | 34B | 95.1 | 93.6 | 74.3 | 67.1 | 82.5 | 64.1 | 78.6 | 4.8 | 33.4 | 25.1 | 12.2 | 36.4 | 4.0 |
| Llama-3.1 | 70B | 96.1 | 93.2 | 83.3 | 67.8 | 85.1 | 68.6 | 77.2 | 32.9 | 52.2 | 46.0 | 33.3 | 51.7 | 6.7 |
| Phi-3-medium | 14B | 88.7 | 76.5 | 77.4 | 59.3 | 75.5 | 58.9 | 67.0 | 17.1 | 23.9 | 22.4 | 26.6 | 36.0 | 5.2 |
| Qwen2.5 | 7B | 83.3 | 81.7 | 73.2 | 57.0 | 73.8 | 53.1 | 75.8 | 17.7 | 31.2 | 28.4 | 28.1 | 39.1 | 5.8 |
| *Specialized Long-Context Optimized Models* | | | | | | | | | | | | | | |
| FILM | 7B | 81.7 | 92.8 | 64.9 | 63.0 | 75.6 | 52.6 | 78.0 | 6.4 | 28.0 | 26.9 | 22.1 | 35.7 | 4.9 |
| ProLong-512k | 8B | 98.5 | 97.8 | 69.4 | 65.5 | 82.8 | 67.2 | 76.4 | 14.4 | 39.1 | 36.7 | 25.9 | 43.3 | 5.9 |
| ChatQA-2 | 8B | 97.1 | **98.1** | 66.8 | 53.6 | 78.9 | 63.2 | **81.3** | 2.9 | 23.7 | 36.2 | 13.9 | 36.9 | 3.7 |
| SEALONG | 8B | 98.4 | 91.0 | 66.6 | 66.1 | 80.5 | 64.9 | 78.5 | 19.6 | **45.0** | 36.2 | 30.1 | 45.7 | 6.6 |
| Mistral | 7B | 72.6 | 74.4 | 64.4 | 52.2 | 65.9 | 47.1 | 63.6 | 8.2 | 25.0 | 19.2 | 20.3 | 30.6 | 4.5 |
| + WildLong | 7B | 95.2 | 95.9 | 67.0 | 64.2 | 80.6 | 62.1 | 74.6 | 12.4 | 34.3 | 34.4 | 29.2 | 41.2 | 6.3 |
| Llama 3.1 | 8B | 98.1 | 91.6 | 66.2 | 66.1 | 80.5 | 66.1 | 77.4 | 18.5 | 39.0 | 37.1 | 28.0 | 44.5 | 6.2 |
| + WildLong | 8B | **98.7** | 95.7 | **74.3** | **67.9** | **84.1** | **67.6** | 78.8 | **22.6** | 40.8 | **38.5** | 30.8 | **46.5** | **6.8** |

long-context QA pairs. SEALONG (Li et al., 2024d) applies preference optimization on Llama-3.1-8B-Instruct with extended-context QA pairs, while ProLong (Gao et al., 2024b) continue-pretrain Llama-3-8B-Instruct to 512K context window and finetune with short-context data.

**Prior Long-Context SFT Data.** We fine-tune Llama-3.1 on open-source long-context instruction-tuning datasets. LongAlpaca (Chen et al., 2024b) covers tasks such as book questions and summarization. LongAlign (Bai et al., 2024) includes QA pairs generated by Claude 2.1 from extended documents, while LongReward (Zhang et al., 2024a) similarly uses GLM4 to produce long-context QA pairs via a self-instruct framework.

### 3.3 EVALUATION BENCHMARKS

We evaluate our model on both long-context and short-context tasks. For long-context tasks, we benchmark against established baselines, while for short-context tasks, we compare performance with the orginal model used for fine-tuning. We assess long-context capabilities using three benchmarks covering a range of input lengths:

**RULER** (Hsieh et al., 2024). This benchmark evaluates four synthetic task types across input lengths ranging from 4K to 128K tokens, including Needle-in-a-haystack (NIAH) retrieval, Multi-hop Tracing with Variable Tracking (VT), Aggregation (Agg), and Question Answering (QA).

**HELMET** (Yen et al., 2025). We test on six HELMET tasks—Retrieval-augmented generation (RAG), Generation with citations (Cite), Passage re-ranking (Rank), Long-document QA (QA), Summarization (Summ), and Many-short in-context learning (ICL). The Recall task is excluded due to overlap with RULER's synthetic NIAH.

**Longbench-Chat** (Bai et al., 2024): Measures instruction-following over long contexts (10K–100K tokens) using 40 English and 10 Chinese real-world queries. `GPT-4-128K` acts as an impartial evaluator.

For short-context tasks, we assess general language understanding and reasoning using MMLU (Hendrycks et al., 2021), Winogrande (Sakaguchi et al., 2020), ARC-C (Clark et al., 2018), and

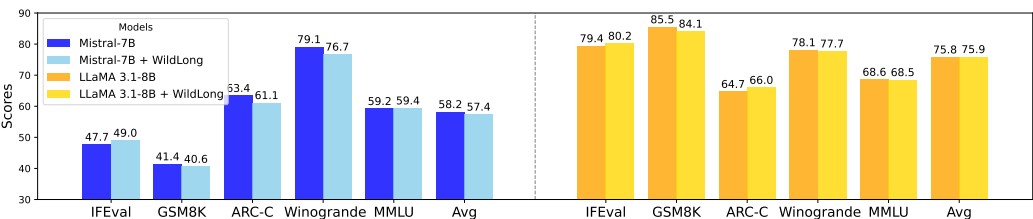

Figure 3: Comparison of short-context performances between finetuned and the baseline models.

Table 2: Comparison of models finetuned with WildLong and other long-context instruction-tuning data. We use 10K data for all dataset.

| Model | Dataset | RAG | ICL | Cite | Rerank | QA | Summ | Avg |
|---|---|---|---|---|---|---|---|---|
| Mistral | w/ LongAlign | 55.1 | 77.3 | 4.3 | 32.6 | 10.6 | 21.6 | 33.6 |
| | w/ LongAlpaca | 56.9 | 69.6 | 4.9 | 32.6 | 27.3 | 19.7 | 35.2 |
| | w/ LongReward | 57.4 | 74.1 | 7.3 | 29.2 | 23.8 | 21.8 | 35.6 |
| | w/ WildLong | 58.1 | 74.1 | 7.2 | 32.1 | 27.2 | 22.0 | **36.8** |
| Llama | w/ LongAlign | 66.1 | 78.0 | 18.3 | 42.6 | 37.4 | 26.4 | 44.8 |
| | w/ LongAlpaca | 66.3 | 79.6 | 14.9 | 38.0 | 38.3 | 25.3 | 43.7 |
| | w/ LongReward | 66.0 | 77.6 | 18.1 | 39.1 | 37.7 | 26.3 | 44.1 |
| | w/ WildLong | 66.0 | 78.6 | 21.4 | 41.5 | 37.6 | 26.5 | **45.3** |

GSM8K (Cobbe et al., 2021), and evaluate instruction-following capabilities with IFEval[9](Zhou et al., 2023).

## 3.4 RESULTS

**Our finetuned models demonstrates strong performance over established models.** We significantly improve upon our baseline models, with Mistral-7B gaining +14.7 and +10.6 points on RULER and HELMET, and Llama-3.1-8B gaining +3.6 and +2.0. Against open-source long-context models, our Llama-3.1-8B matches or exceeds larger alternatives. Notably on LongBench-Chat, our Llama-3.1-8B model outperforms most established models except for proprietary ones. We also outperform specialized long-context methods. Despite using ten times more data, FILM scores lower than our Mistral-7B (e.g., 75.6 vs. 80.6 on RULER). SEALONG, based on Llama-3.1-8B-Instruct achieves lower scores, with an 8-point deficit on RULER compared with our Llama-based model. ProLong and ChatQA-2 perform well on synthetic tasks but struggle with real-world queries and complex tasks. These results highlight the effectiveness of our framework.

**Our method enhances performance compared to other long-context instruction tuning data.** We compare WildLong with existing long-context instruction tuning datasets, LongAlign, LongAlpaca and LongReward, using 10K samples from each. As shown in Table 2, WildLong achieves the highest average HELMET scores on both Mistral-7B (36.8) and Llama-3.1-8B (45.3) To verify that these gains stem from our novel instruction synthesis framework rather than the use of GPT-4 as a teacher model, we conducted an additional ablation study detailed in Appendix C.3. It performs well across both information extraction tasks (e.g., RAG, QA) and generation tasks (e.g., citation, summarization), indicating a stronger ability to handle the diverse cognitive demands of long-context reasoning. We analyze the distribution of tasks covered by WildLong in Appendix E. These results highlight WildLong's effectiveness in generating data that support both focused retrieval and broad integration, improving task generalization. Beyond performance, scalability is a key advantage. Existing datasets are typically capped at under 12K samples, whereas WildLong scales efficiently to much larger volumes. We examine this scalability and its impact in Section 3.5.

**Short context performance is preserved without mixing short-context data.** Previous works (An et al., 2024b; Bai et al., 2024; Zhang et al., 2024a) mix short-context instruction-tuning data into the finetuning data to mitigate degradation in short-context capabilities after long-context alignment.

---

[9]Details on evaluation settings and results on LongBench v2 (Bai et al., 2025) are in Appendix B.4 and D.

Table 3: Effect of graph-based modeling adopted by WildLong compared with two baseline methods.

| Model | Dataset | RAG | ICL | Cite | Rerank | QA | Summ | Avg |
|---|---|---|---|---|---|---|---|---|
| Mistral | w/ WildChat-long | 56.8 | 72.6 | 3.1 | 30.0 | 25.9 | 16.2 | 34.1 |
| | w/ Simple-Instruct | 57.5 | 75.0 | 3.5 | 29.4 | 26.4 | 17.2 | 34.8 |
| | w/ WildLong | 58.7 | 73.1 | 10.9 | 34.9 | 27.5 | 22.3 | **37.9** |
| Llama | w/ WildChat-long | 65.7 | 77.8 | 17.3 | 40.9 | 37.8 | 26.3 | 44.3 |
| | w/ Simple-Instruct | 67.5 | 77.5 | 18.3 | 41.6 | 37.1 | 25.7 | 44.6 |
| | w/ WildLong | 66.3 | 78.0 | 20.4 | 42.5 | 38.0 | 26.7 | **45.3** |

In contrast, our approach exclusively employs long-context data while effectively preserving short-context performance. Referring to Figure 3, we maintain an average score of 75.9 for Llama-3.1-8B, comparable to the baseline 75.8. For Mistral-7B, we observe a slight drop of less than one point, potentially due to changes in RoPE base. We explain this further in Section 3.5. These results underscore the effectiveness of our dataset: finetuning on general, realistic long-context data significantly enhances long-context capabilities while largely preserving short-context performance without additional data mixing.

## 3.5 ABLATION STUDIES

**Effectiveness of graph-based modeling.** To evaluate our graph-based instruction generation, we compare it with two baselines, each using 20k samples. Simple-Instruct extracts instructions from WildChat and pairs them with SlimPajama documents, while WildChat-Long fine-tunes on a filtered long-context subset of WildChat. We fine-tune both Mistral-7B-Instruct-v0.2 and Llama-3.1-8B-Instruct on these datasets and evaluate on HELMET. As shown in Table 3, our graph-based method consistently outperforms the baselines. For example, Mistral-7B achieves 37.9, outperforming WildChat-Long and Simple-Instruct by +3.8 and +3.1 points. These gains—especially in citation, reranking, and summarization—highlight the graph-based method's ability to generate diverse and challenging instructions while preserving generalizability. Scalability is another key advantage. The baselines rely on human-chatbot data or heuristics, limiting scale due to cost and manual effort. In contrast, WildLong scales efficiently via graph-based modeling and adaptive generation. Section 3.5 shows continued performance gains with more data, validating the scalability of our approach.

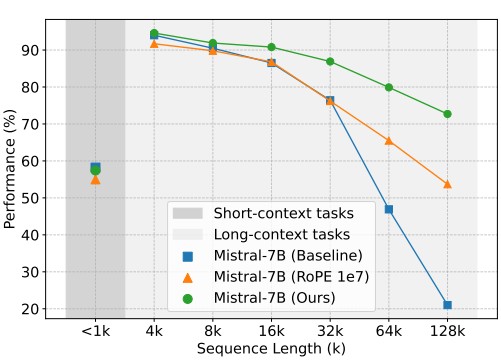

Figure 4: Length-wise performance of Mistral variants on short- and long-context tasks.

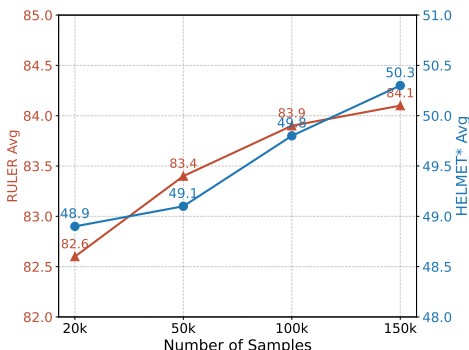

Figure 5: Llama performance on RULER and HELMET* with increasing synthetic data. HELMET* excludes tasks requiring model-based evaluation.

**Effectiveness of WildLong under RoPE scaling.** We investigate the impact of RoPE scaling on Mistral-7B, comparing three variants: (1) Mistral-7B (Baseline): The original model with context length 32k and RoPE base $1e6$, (2) Mistral-7B (RoPE $1e7$): Extended RoPE base of $1e7$, and (3) Mistral-7B (Ours): RoPE base $1e7$, finetuned with our WildLong data. Performance is evaluated on short-context tasks ($<$1k) and long-context tasks (RULER, 4k-128k). The length-wise performance is shown in Figure 4. Our results reveal that increasing the RoPE base enables support for longer contexts, with gains of +18.6 at 64K and +32.7 at 128K over the baseline. However, this comes with a significant trade-off, as short-context performance drops markedly from 58.2 to 55.0, and

mid-range (4k-8k) performance slightly declines. Finetuning with WildLong mitigates these trade-offs, recovering short-context performance to 57.4 while further boosting mid- and long-context performance. These results highlight the inherent trade-off in RoPE scaling. Finetuning with generalized long-context datasets, such as Wildlong, mitigates this issue, improving both mid-range and long-context.

**Effectiveness of scaling data size.** To assess the impact of scaling synthetic data, we fine-tuned Llama-3.1-8B-instruct on varying sizes of WildLong dataset (20k, 50k, 100k, and 150k samples). Figure 5 shows steady performance gains on the RULER ($82.6 \rightarrow 84.1$) and a subset of HEL-MET benchmarks ($48.9 \rightarrow 50.3$), excluding model-based evaluation tasks for efficiency. The non-saturating performance at 150K samples validates our WildLong framework's ability to generate high-quality data, achieving consistent gains at a scale an order of magnitude larger than prior datasets. Identifying the precise saturation point, which is fundamentally tied to model capacity, was beyond our scope due to resource constraints. Our framework thus enables the full exploration of these scaling laws with larger models as a key direction for future work.

## 4 RELATED WORK

**Long-context extension of LLMs.** Many works extend LLM context windows with minimal training, using position extrapolation (Chen et al., 2023; Peng et al., 2024b; Su et al., 2021; Ding et al., 2024; Chen et al., 2024a; Liu et al., 2024a; Zhu et al., 2024; Wu et al., 2024; Hu et al., 2024) or modifying attention mechanisms (Jin et al., 2024; Xiao et al., 2024b;a; Ding et al., 2023; An et al., 2024a; 2025). Other approaches propose architectural innovations for efficient long-context modeling (Lieber et al., 2024; Bertsch et al., 2024; Wang et al., 2024b; Yen et al., 2024). Large-scale solutions leverage continued pretraining or supervised finetuning on long-context data (Dubey et al., 2024; GLM et al., 2024), though such methods are often costly and resource-intensive. To mitigate this, synthetic long-context datasets have emerged. An et al. (2024b) generate QA for short contexts and concatenate them, while Zhao et al. (2024a) construct synthetic tables to enhance long-context reasoning. Xu et al. (2025a) build contexts by combining semantically related paragraphs from NarrativeQA. Structured datasets have also targeted specific tasks: Chen et al. (2024c) model document correlations for multi-hop QA generation; Bai et al. (2024) use Self-Instruct for long-context instruction synthesis but limit prompts to four task types; Xiong et al. (2025a) focus on synthetic key-value retrieval for multi-document reasoning. While promising, these efforts often remain narrow in scope or require heavy manual or computational effort.

**Scaling synthetic data creation.** Previous alignment datasets relied on human interactions with LLMs (Conover et al., 2023; Zhao et al., 2024b; Zheng et al., 2024; Köpf et al., 2023), but manual instruction crafting is labor-intensive. Recent approaches scale synthetic instruction datasets by prompting LLMs using a small set of human-annotated seeds (Yu et al., 2024; Wang et al., 2023; Taori et al., 2023; Xu et al., 2024a; Sun et al., 2024). Keypoint-driven methods (Li et al., 2024a; Tang et al., 2024; Huang et al., 2024) enhance diversity using topical cues or knowledge bases, while PersonaHub (Ge et al., 2024) introduces billions of personas to maximize coverage. Our approach shares the high-level idea of keypoint-guided generation but focuses on realistic, document-grounded long-context instructions derived from real-world conversational data. By incorporating document-type–specific meta-information, our framework enables scalable and diverse long-context data generation with minimal manual effort.

## 5 CONCLUSION

We propose WildLong, a framework for synthesizing diverse, scalable, and realistic instruction-response datasets for long-context tasks. It combines meta-information extraction, graph-based modeling, and adaptive instruction generation to produce high-quality, context-aware instructions. Our fine-tuned models consistently outperform prior long-context–optimized baselines across benchmarks. Notably, they retain strong short-context performance without mixing in short-context data. Analysis shows that WildLong enables models to generalize well across task types and exhibit greater robustness than existing approaches. Moreover, it scales effectively, with continued performance gains as data size increases. WildLong offers a practical path toward robust long-context LLMs and contributes to broader efforts in generalizing synthetic data for complex instruction tuning.

ETHICS STATEMENT

This research was conducted in accordance with the principles outlined in the ICLR Code of Ethics. Our work aims to contribute to society and human well-being by improving the long-context reasoning capabilities of language models, which can benefit complex, information-intensive applications in fields like academic research.

In line with upholding high standards of scientific excellence and transparency, we have provided a detailed description of our data generation framework, training procedures, and evaluation settings (Sections 2 and 3). All prompts used in the WildLong framework are disclosed in Appendix I to ensure the reproducibility of our results. We respect the work required to produce new ideas and artifacts by using publicly available datasets (SlimPajama and WildChat) for our research and have properly credited their creators while adhering to their licenses, as detailed in Appendix K.

We have taken measures to avoid harm. We acknowledge that synthetic data generation can inherit and amplify biases from the source data and the generator model. To mitigate the risk of creating harmful or toxic content, all data was generated using the Azure OpenAI service, which incorporates content moderation filters to reject unsafe outputs. We further discuss the potential for misuse, such as the creation of coherent disinformation, in our Broader Impact statement in Appendix L, reflecting our commitment to being honest and transparent about the limitations and potential negative consequences of our work. Our research does not involve human subjects, and all data used is from pre-existing, public sources.

REPRODUCIBILITY STATEMENT

To ensure the reproducibility of our research, we provide comprehensive details of our methodology and experiments. The WildLong framework is fully described in Section 2, with the exact prompts used for data generation available in Appendix I. All training hyperparameters, model specifics, and computational infrastructure are detailed in Section 3 and Appendix B.5, while evaluation settings are specified in Appendix B.4. Our data sources, processing steps, and dataset analysis are described in Section 2 and Appendices B, E, and K. We plan to release our source code, the synthesized WildLong dataset, and all fine-tuned model checkpoints upon publication to facilitate further research.

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

## A    THE USE OF LLMS

We utilized an LLM as a writing assistant to improve the clarity, grammar, and overall structure of this manuscript. Its use included rephrasing sentences for conciseness, ensuring a formal academic tone, and restructuring content, such as converting bulleted analyses into paragraphs. The LLM's role was strictly confined to editing and polishing the original text.

## B    EXPERIMENTAL DETAILS

Here we list additional experimental details for our implementation and experiments.

### B.1    DOCUMENT TYPE CLUSTERING

Document types are initially extracted as free-form values during meta-information extraction. To group them into coherent categories, we apply K-Means clustering, setting the number of clusters to 10 to balance generalization and specificity given the dataset diversity. Each cluster is then assigned a consolidated document type label by rewriting its center. Figure 6 shows the distribution of these categories.

### B.2    DOCUMENT CLASSIFIER

We trained a random forest classifier on semantic features extracted from a small language model. The classifier was trained on annotations of 20,000 long documents from SlimPajama, achieving 90% accuracy on a held-out test set. Specifically, we annotated $20,000$ long documents sampled from SlimPajama using GPT-4. The annotation prompt explicitly required the output to match one of the predefined document types, ensuring consistency with the categories defined during meta information clustering. Using these annotations, we trained a random forest classifier on semantic features extracted with StableLM-2-1.6B (Bellagente et al., 2024), where the mean of the last layer's hidden states was used as the feature representation. The classifier achieved 90% accuracy on a held-out test set, enabling efficient and accurate predictions of document types for unseen SlimPajama data.

### B.3    INSTRUCTION GENERATION WITH PATHS

To synthesize instructions aligned with sampled meta information paths, we prompt GPT-4 with a one-shot demonstration derived from seed paths extracted from the WildChat long conversations. Each seed path includes all meta information fields and a corresponding simplified instruction. Given a sampled path $\hat{P}$, we identify the most similar seed path $P^*$ based on the of their nodes. The similarity between paths is computed as intersection_sim$(\hat{P}, P^*) = |\hat{P} \cap P^*|$. The selected example path and its instruction are included in the prompt to guide GPT-4 in generating a new instruction given a new path. This ensures the generated instruction aligns with the sampled meta information criteria, while benefiting from the contextual relevance provided by the seed example. GPT-4 synthesizes a natural language instructions adhering to the sampled path's constraints with the prompt shwon in Table 12.

### B.4    EVALUATION SETTINGS

For short-context evaluation, we utilize the lm-evaluaton-harness framework Gao et al. (2024a) and following the evaluation settings in (Beeching et al., 2023): 25-shots for ARC-C, and 5-shots for MMLU, Winogrande and GSM8K. We use 0-shot for IFEval. We report the acc_norm metric for ARC-C, the acc metric for MMLU, Winogrande and GSM8K. We average the metrics prompt_level_strict_acc, inst_level_strict_acc, prompt_level_loose_acc, and inst_level_loose_acc for IFEval.

For long-context evaluations, we evaluate our models and all baselines following the settings in the original benchmarks. Table 4 presents the sources of evaluation results for the models across three benchmarks.

Table 4: Evaluation source for each model on three benchmarks. ✓ indicates that the evaluation was conducted by ourselves, while ★ indicates that results were sourced from the original benchmark.

| Models | RULER | HELMET | Longbench-Chat |
|---|---|---|---|
| *Proprietary Long-Context Models* | | | |
| Gemini-1.5-Pro | ★ | ★ | ✓ |
| GPT-4 | ★ | ★ | ★ |
| *Open-Sourced Pretrained Long-Context Models* | | | |
| GLM-4-1M | ★ | ✓ | ✓ |
| Yi-200k | ★ | ★ | ✓ |
| Llama-3.1-70B | ★ | ★ | ✓ |
| Phi-3-medium | ★ | ★ | ✓ |
| Qwen2.5 | ✓ | ✓ | ✓ |
| Mistral-7B | ✓ | ✓ | ✓ |
| Llama-3.1-8B | ✓ | ✓ | ✓ |
| *Specialized Long-Context Optimized Models* | | | |
| FILM | ✓ | ✓ | ✓ |
| ProLong-512k | ✓ | ✓ | ✓ |
| ChatQA-2 | ✓ | ✓ | ✓ |
| SEALONG | ✓ | ✓ | ✓ |

## B.5 TECHNICAL DETAILS

We employ several open-source libraries and tools for model training. Specifically, we use PyTorch (Paszke et al., 2019) and the Hugging Face Transformers library (Wolf, 2019) for implementing and fine-tuning the model. To enhance computational efficiency, we integrate FlashAttention 2 (Dao, 2024) for optimized attention computation. The fine-tuning process is conducted on eight AMD Radeon Instinct MI300 GPUs, each equipped with 192GB of memory. Training on 150K synthetic data samples requires approximately 480 GPU hours.

## C ADDITIONAL ABLATION STUDIES

### C.1 ABLATION STUDY ON PATH SAMPLING STRATEGY

In our framework, the goal of the Meta Information Path Sampling stage (Section 2.3) is to generate instruction templates that are both realistic, reflecting real-world co-occurrence patterns, and diverse, ensuring broad coverage to enhance model generalization. We selected a weighted random walk as it provides a principled method for balancing these two objectives. The edge weights, derived from meta-information frequencies in the WildChat dataset, bias sampling toward plausible combinations, ensuring realism. Simultaneously, the probabilistic nature of the walk allows the framework to explore less common but still valid paths, preventing deterministic generation of only the most frequent instructions and thereby fostering diversity.

To empirically validate our choice, we conducted an ablation study comparing our method against two baselines designed to prioritize one objective over the other. The first baseline, **Greedy Max-Weight Sampling**, prioritizes realism by deterministically selecting the neighbor with the highest edge weight at each step. The second, **Uniform Random Sampling**, prioritizes diversity by ignoring all edge weights and selecting a neighbor from a new meta-information category uniformly at random. For each of the three strategies, we generated a dataset of 10,000 samples and fine-tuned the Llama-3.1-8B-Instruct model. Due to computational constraints, we evaluated performance on the four HELMET tasks that do not require model-based evaluation: Retrieval-Augmented Generation (RAG), Many-short In-Context Learning (ICL), Generation with Citations (Cite), and Passage Re-ranking (Rank).

Table 5: Performance comparison of different path sampling strategies on a subset of HELMET.

| Sampling Strategy | RAG | ICL | Cite | Rerank | Avg |
|---|---|---|---|---|---|
| Greedy Max-Weight | 64.8 | 77.7 | 17.7 | 37.9 | 49.5 |
| Uniform Random | 64.7 | 76.7 | 17.6 | 38.9 | 49.5 |
| **Weighted Random (Ours)** | **66.0** | **78.6** | **21.4** | **41.5** | **51.9** |

**Analysis**    The results in Table 5 provide clear empirical validation for our approach. Our proposed weighted random walk (Avg: 51.9) significantly outperforms both the Greedy Max-Weight and Uniform Random baselines, which perform identically on average (49.5). The lower performance of the greedy baseline suggests that over-optimizing for realism at the expense of diversity creates a less effective training set. Conversely, the uniform baseline's result demonstrates that completely ignoring the realistic data distributions is equally detrimental. Our weighted random walk successfully strikes the essential balance between these two competing priorities, leading to a demonstrably superior dataset for instruction tuning.

## C.2    Ablation Study on Path Length

The path length, $N$, represents the number of meta-information fields sampled to construct an instruction template. This hyperparameter governs the trade-off between instruction simplicity and specificity. A path that is too short may result in generic instructions, while one that is too long may produce overly constrained or convoluted requests. To empirically justify our choice of $N = 6$, we conducted an ablation study to evaluate its effectiveness against alternative lengths.

**Experimental Setup**    We evaluated three distinct path lengths. In addition to our chosen value of $N = 6$, we tested $N = 3$ to represent simpler, less constrained instructions, and $N = 9$ to represent more complex and potentially unnatural instructions. For each setting, we generated a 10,000-sample dataset and fine-tuned the Llama-3.1-8B-Instruct model, holding all other variables constant. We report the performance on the same subset of HELMET tasks used in our prior ablation study.

Table 6: Performance comparison for different path lengths ($N$) on a subset of HELMET. The results confirm that $N = 6$ achieves the best trade-off between task richness and coherence.

| Path Length (N) | RAG | ICL | Cite | Rerank | Avg |
|---|---|---|---|---|---|
| N = 3 | 65.7 | 77.6 | 17.5 | 41.4 | 50.5 |
| N = 9 | 64.7 | 76.7 | 17.5 | 38.8 | 49.4 |
| **N = 6 (Ours)** | **66.0** | **78.6** | **21.4** | **41.5** | **51.9** |

**Analysis**    The results presented in Table 6 provide strong empirical support for our selection of $N = 6$. Our main model (Avg: 51.9) significantly outperforms both the $N = 3$ (Avg: 50.5) and $N = 9$ (Avg: 49.4) variants. These findings align with our initial hypothesis: shorter paths ($N = 3$) likely generate overly generic tasks that do not challenge the model sufficiently, while longer paths ($N = 9$) produce overly specific or rigid instructions that are less natural and may hinder generalization. Therefore, a path length of $N = 6$ strikes the most effective balance, yielding a dataset with sufficient complexity to be challenging yet coherent enough to be effective for instruction tuning.

## C.3    Ablation Study on the Impact of the Teacher Model

A potential confounding factor in our main results is the use of GPT-4 as the teacher model for generating instruction-response pairs, as other baseline datasets do not rely on a proprietary model of this scale. To isolate the contribution of our WildLong instruction synthesis framework from the quality of the teacher model, we conducted an ablation study.

**Experimental Setup** We replicated our data generation process using a capable but modest-sized open-source model, **Qwen2.5-14B-Instruct-1M**, as the teacher. This model was used to generate the adapted instructions and corresponding responses for 10,000 instruction templates created by our framework. We then fine-tuned the Llama-3.1-8B-Instruct model on this new dataset (referred to as "w/ WildLong (Qwen Teacher)"). We report the performance on the same subset of HELMET tasks as in the path sampling strategy ablation. The results are compared against the baseline datasets and our original GPT-4-annotated data in Table 7.

Table 7: Performance comparison on a subset of HELMET tasks to evaluate the impact of the teacher model. All models are fine-tuned on Llama-3.1-8B-Instruct using 10k data samples. Our WildLong framework with an open-source teacher model still outperforms all baseline datasets, demonstrating that the gains are primarily driven by the instruction synthesis method.

| Dataset | RAG | ICL | Cite | Rerank | Avg |
|---|---|---|---|---|---|
| w/ LongAlign | 66.1 | 78.0 | 18.3 | 42.6 | 51.2 |
| w/ LongAlpaca | 66.3 | 79.6 | 14.9 | 38.9 | 49.9 |
| w/ LongReward | 66.0 | 77.6 | 18.1 | 39.1 | 50.2 |
| w/ WildLong (Qwen Teacher) | 65.9 | 77.8 | 20.7 | 42.1 | **51.6** |
| w/ WildLong (GPT-4 Teacher) | 66.0 | 78.6 | 21.4 | 41.5 | **51.9** |

**Analysis** The analysis of our results reveals two key insights. First and foremost, the model trained with the Qwen teacher achieves an average score of 51.6, a result that surpasses all baseline datasets. This strongly indicates that the primary driver of performance gains is the high quality and diversity of instructions generated by the WildLong framework, independent of the teacher model used for response generation. Secondly, the minimal performance gap between the model trained with the Qwen teacher (51.6) and the one with the GPT-4 teacher (51.9) demonstrates the robustness of our framework. This finding confirms that while a more powerful teacher offers a slight edge, our method is capable of producing a competitive and high-quality dataset even with more accessible, open-source models.

This experiment validates that the novelty and effectiveness of our work lie in the meta-information-driven, graph-based instruction synthesis process. While the choice of teacher model can refine response quality, it is not the principal reason for the superior performance of our dataset.

# D ADDITIONAL RESULTS

We present further experimental results in this section.

## D.1 PERFORMANCE ON LONGBENCH V2 BENCHMARK

We provide a supplementary evaluation on LongBench v2 to further test our models on a challenging set of long-context tasks. While its smaller scale, comprising 503 instances, and its inclusion of context lengths up to 2M tokens placed it outside our primary evaluation suite, these results offer further validation of WildLong's effectiveness. As shown in Table 8, fine-tuning with our dataset yields substantial improvements, highlighting the quality and effectiveness of our data synthesis framework. For the Mistral-7B evaluation, we adjusted the RoPE base from 1e6 to 1e7 to support longer context windows, consistent with the settings in the main text.

**Analysis** The results clearly demonstrate the value of fine-tuning with the WildLong dataset. Both Mistral-7B (+3.5) and Llama-3.1-8B (+2.4) achieve marked improvements in their overall scores, confirming the broad-based benefits of our data. The gains are particularly pronounced in long-context reasoning; for instance, Mistral-7B improves substantially on the Medium (+7.0) and Long (+3.7) splits, directly validating our data's effectiveness in this domain. Furthermore, the consistent score increases across both Easy and Hard subsets indicate that WildLong's instructional diversity prepares models for a wide range of reasoning challenges.

Table 8: Performance on the LongBench v2 benchmark. Models fine-tuned with our WildLong dataset show consistent improvements across overall, difficulty-based, and length-based splits.

| Model | Overall | Easy | Hard | Short | Medium | Long |
|---|---|---|---|---|---|---|
| Mistral-7B | 24.7 | 27.1 | 23.2 | **27.2** | 22.8 | 24.1 |
| w/ WildLong | **28.2** | **30.7** | **26.7** | 26.7 | **29.8** | **27.8** |
| Llama-3.1-8B | 28.2 | 29.2 | 27.7 | 32.2 | 26.5 | 25.0 |
| w/ WildLong | **30.6** | **31.2** | **30.2** | **36.7** | **27.4** | **26.9** |

Table 9: Performance comparison among single-document, multi-document, and a mixture of single- and multi-document data on HELMET.

| Model | Dataset | RAG | ICL | Cite | Rerank | QA | Summ | Avg |
|---|---|---|---|---|---|---|---|---|
| | w/ Single | 59.7 | 74.7 | 9.7 | 33.4 | 26.9 | 21.8 | 37.7 |
| Mistral | w/ Multi | 56.9 | 72.6 | 12.3 | 33.6 | 27.9 | 23.8 | 37.9 |
| | w/ WildLong | 58.7 | 73.1 | 10.9 | 34.9 | 27.5 | 22.3 | 37.9 |
| | w/ Single | 66.3 | 78.2 | 20.0 | 42.7 | 38.2 | 26.1 | 45.2 |
| Llama | w/ Multi | 66.0 | 77.8 | 20.3 | 43.1 | 37.5 | 26.9 | 45.3 |
| | w/ WildLong | 66.3 | 78.0 | 20.4 | 42.5 | 38.0 | 26.7 | 45.3 |

## D.2 EFFECTIVENESS OF MULTI-DOCUMENT DATA.

We evaluate the impact of multi-document supervision by fine-tuning Mistral and Llama models on 20k samples under three settings: single-document, multi-document, and a mixture of both (WildLong). Our results, presented in Table 9 and Table 10, reveal a clear trade-off between different data compositions. Single-document data consistently performs better on tasks requiring focused information extraction, such as RAG and QA. In contrast, multi-document data is more effective for complex reasoning tasks that involve integrating information across multiple sources. This is evidenced by notable performance gains in citation generation (Cite) and summarization (Summ). For instance, the Mistral model's Cite score improves from 9.7 to 12.3 with multi-document data. Despite these task-specific shifts, the overall average performance on HELMET remains stable, with slight improvements for both Mistral ($37.7 \rightarrow 37.9$) and Llama ($45.2 \rightarrow 45.3$) when trained on multi-document data. We observe a similar trend on the RULER benchmark (Table 6), where multi-document data better supports complex reasoning tasks like variable tracking (VT). The mixed setting (WildLong) performs on par with multi-document data in terms of average performance on HELMET, but offers more balanced gains across task types. These results suggest that the optimal ratio of single- to multi-document data is a hyperparameter that likely depends on the target model and downstream applications. The 2:1 ratio used in our main WildLong dataset offers a balanced, general-purpose mixture. This highlights the flexibility of the WildLong framework to enable more targeted studies into data composition. For instance, a promising future research direction is automated ratio tuning, which involves developing methods to find the optimal data mixture for a specific base model. Another avenue is curriculum learning, where one could explore strategies that dynamically adjust the proportion of single- and multi-document data during fine-tuning to maximize learning efficiency.

## E DATASET ANALYSIS

We analyze the distribution of document types and task types in our dataset, as shown in Figure 6. Following Xu et al. (2025b), we utilize Llama-3.1-8B-Instruct to categorize task types by prompting it with instructions generated by the WildLong framework. The model is explicitly instructed to label the task category, ensuring a systematic and consistent classification.

Table 10: Performance comparison among single-document, multi-document, and a mixture of single-and multi-document data on RULER.

| Model | Dataset | NIAH | VT | Agg | QA | Avg |
|-------|---------|------|-----|-----|-----|-----|
| Mistral | w/ Single | 91.6 | 90.9 | 63.9 | 64.2 | 77.7 |
| | w/ Multi | 92.1 | 94.4 | 66.9 | 64.1 | 79.4 |
| | w/ WildLong | 91.4 | 92.0 | 64.7 | 63.9 | 78.0 |
| Llama | w/ Single | 98.6 | 93.0 | 70.5 | 68.2 | 82.6 |
| | w/ Multi | 98.8 | 93.0 | 69.3 | 67.0 | 82.0 |
| | w/ WildLong | 98.9 | 93.7 | 70.0 | 67.7 | 82.6 |

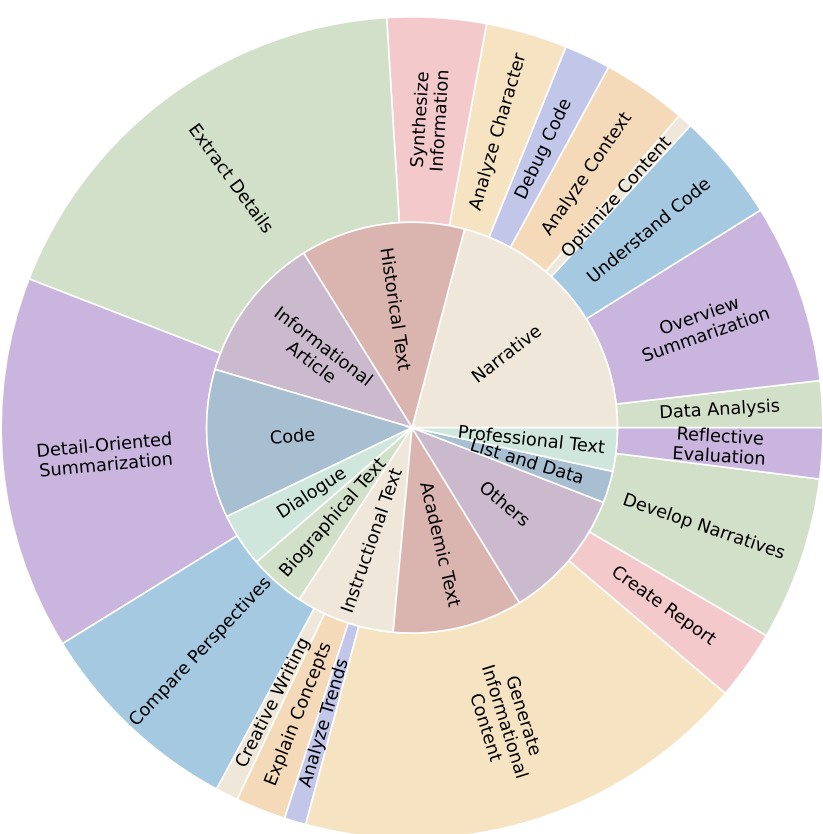

Figure 6: Distribution of document types (inner circle) and task types (outer circle) in our dataset.

# F  DATASET VALIDATION AND QUALITY CONTROL

To address the validity of our synthetic dataset, we provide a quantitative analysis of how WildLong instructions align with real-world user needs and an overview of our quality control process for generated responses.

## F.1  ALIGNMENT WITH REAL-WORLD USER NEEDS

We conducted a comparative analysis to ensure the instruction distribution in our synthetic WildLong dataset quantitatively mirrors the real long-context user-chatbot conversations from which it was derived (WildChat).

**Methodology**  We randomly sampled 2,000 instructions from the finalized WildLong dataset and 2,000 long-context instructions from the WildChat dataset. To ensure consistent and unbiased classification, we prompted Llama-3.1-8B-Instruct to categorize each instruction into one of 25 predefined task types.

**Results**  Our analysis reveals a strong correspondence between the synthetic and real-world distributions, evidenced by a Pearson correlation of 0.89. This high correlation reflects the preservation of core user needs, as the top three tasks in WildChat—Extract Details, Generate Informational Content, and Detail-Oriented Summarization—are identically ranked in WildLong. The minor distributional shifts are intentional, resulting from our framework's deliberate sampling of underrepresented but critical reasoning tasks like multi-document Compare Perspectives. These findings validate that WildLong faithfully reflects the distribution of real user needs while strategically expanding its diversity to better support complex reasoning capabilities.

## F.2  RESPONSE QUALITY CONTROL AND ERROR ANALYSIS

Ensuring the quality of generated responses (i.e., factual consistency, low hallucination, and clarity) was a primary focus during dataset construction. We implemented a multi-stage quality control process.

**Iterative Prompt Refinement**  For each instruction-response generation task that utilized GPT-4, we employed a structured manual verification process to finalize the prompt. This process involved sampling and manually evaluating 100 responses for each candidate prompt, assessing them for coherence, contextual relevance, and adherence to constraints. A prompt was approved for large-scale generation only after at least 90% of its sampled responses met all quality criteria. If a prompt failed to meet this threshold, it was refined and the verification process was repeated. This iterative cycle, typically performed 3–8 times per task, allowed us to systematically identify and mitigate common failure modes like hallucination and ambiguity before commencing full-scale data generation

**Final Dataset Audit**  In addition to the prompt-level validation, we conducted a small-scale manual audit on a random sample of 200 final instruction-response pairs from the WildLong dataset. The audit confirmed the high quality of the data, revealing that 97.5% of responses were factually consistent with the provided context, with the few inconsistencies being minor. Furthermore, the hallucination rate was exceptionally low at 1%, with these instances typically being minor stylistic embellishments rather than significant factual fabrications. Similarly, only 1.5% of responses exhibited slight ambiguity but remained generally understandable.

This two-pronged approach, combining proactive prompt engineering with a final quality audit, ensures that the WildLong dataset maintains a high degree of fidelity and reliability.

## F.3  SAFETY ANALYSIS

We employed a multi-stage process to mitigate potential safety risks in the WildLong dataset. During the generation phase, we utilized the Azure OpenAI service, which incorporates a strict content filtering system. This mechanism automatically rejected responses to harmful or sensitive inputs, serving as an initial layer of protection by ensuring such cases were excluded from our raw data.

Following generation, we conducted a comprehensive safety audit on the full dataset using Llama-Guard-2 (Team, 2024), a state-of-the-art model for detecting harmful content. The analysis indicated that the dataset is predominantly safe, with less than 3% of instances flagged as potentially harmful. A majority of these flagged cases (1.9% of the total) fell into the "specialized advice" category, which includes responses offering potentially specialized legal, medical, or financial advice, consistent with the academic nature of many source documents. To finalize the dataset, all instances identified as potentially harmful by Llama-Guard-2 were filtered out and removed.

## G  DISCUSSION ON RoPE BASE

Recent studies demonstrate that adjusting the base value in Rotary Position Embedding (RoPE) is an effective technique for enhancing a language model's ability to handle long-context sequences (bloc97, 2023; Liu et al., 2024c). By increasing the RoPE base parameter (e.g., from $10^4$ to $10^6$), the wavelength of the positional encoding grows exponentially as $\lambda_i \propto \text{base}^{2i/d}$, where $d$ is the embedding dimension. This prolongs the non-repeating positional patterns across distant tokens, mitigating the encoding collisions that can impair long-range dependency modeling. Practical implementations like Code Llama (Grattafiori et al., 2023) and ChatGLM (GLM et al., 2024) have adopted this base scaling to extend their context windows to 16K+ tokens.

However, this architectural modification often introduces a trade-off, where gains in long-context capabilities can lead to a degradation in short-context performance. Our own experiments clearly illustrate this phenomenon. As shown in our ablation study (Section 3.5, Figure 4), simply extending the RoPE base of Mistral-7B from 1e6 to 1e7 resulted in a 3.2-point drop in its average short-context score before any fine-tuning.

Crucially, this analysis highlights the robustness and quality of our `WildLong` dataset. After fine-tuning on `WildLong`, the Mistral-7B model recovered most of this loss, nearly reaching its original baseline performance. This demonstrates that our dataset does not cause the degradation; rather, it effectively mitigates the negative side effects of the RoPE scaling modification. This conclusion is further substantiated by our results with Llama-3.1-8B. This model natively supports a long context and required no RoPE modifications, and consequently, it exhibited no performance degradation on short-context tasks after being fine-tuned on `WildLong`. This confirms that fine-tuning on our general, realistic long-context data is sufficient to preserve short-context capabilities while significantly boosting long-context performance.

## H  EXAMPLES FROM WILDLONG

We present several examples generated by our framework to illustrate the diversity and quality of the synthetic data used for long-context instruction tuning. Each example includes a long context, an instruction, and the corresponding response.

**Example 1**

**Long Context**

[A Review of] "Reflections: Van Eyck and the Pre-Raphaelites," at the National Gallery (2 October 2017-2 April 2018) Madeleine Emerald Thiele MA, School of Art, Aberystwyth University [Victorian Web Home —¿ Visual Arts —¿ Victorian Painting —¿ The Pre-Raphaelites —¿ Reviews] More details of all the illustrations, except for the first one, can be found on our own website. Click on them for larger images, sources and further comments. The first image (below right) is Jan van Eyck's Portrait of a Man (Self Portrait?), 1433 (© National Gallery, NG222, Sunley Room), kindly supplied by the gallery, like several of the others, for purposes of this review. — Jacqueline Banerjee Exhibition Context Jan van Eyck's The Arnolfini Portrait (1454, National Gallery) entered the collection of the National Gallery in 1842. Having spent much of its then four hundred year history in private collections, the painting was put on public display in 1843 for the first time. The Times reported that in that year alone nearly half a million people had visited the painting. ......

**Instruction**

Considering the significant influence of Jan van Eyck's The Arnolfini Portrait on the Pre-Raphaelites as discussed in the document, what are some subtle visual elements or motifs that could be incorporated into a new painting to evoke this historical connection and convey a sense of continuity with the Pre-Raphaelite tradition?

**Response**

Based on the influence of Jan van Eyck's The Arnolfini Portrait on the Pre-Raphaelites, there are several subtle visual elements and motifs that could be incorporated into a new painting to evoke a historical connection and convey a sense of continuity with the Pre-Raphaelite tradition:

1. **Convex Mirror**: Including a convex mirror in the background of the painting can serve as a direct homage to The Arnolfini Portrait. This mirror could reflect elements within the scene or hidden details, creating layers of meaning and dialogue, much like van Eyck's original.

2. **Detailed Interiors**: Like the meticulous domestic setting in The Arnolfini Portrait, the new painting could feature a richly detailed interior with symbolic objects. Items such as prayer beads, religious symbols, or everyday household items can add depth and context to the scene.

3. **Rich Textures and Colors**: Emulate the jewel-like colors and intricate textures characteristic of both van Eyck and the Pre-Raphaelites. Use vibrant greens, deep blues, and detailed fabric textures to create a visually stunning and historically resonant piece.

4. **Symbolic Gestures**: Incorporate subtle hand gestures and body language that hint at deeper narratives, similar to Giovanni's raised hand and the clasped hands of the couple in The Arnolfini Portrait. These gestures can suggest relationships and emotions without overtly stating them.

5. **Mirrored Poses**: Reflect the original painting's composition by positioning figures in a way that mirrors the placement of the Arnolfini couple. This can create a visual link to the historical piece while allowing for new interpretations and stories.

6. **Reflected Figures**: Just as van Eyck included figures reflected in the mirror, you could include reflections of additional characters or the artist themselves, adding layers of interaction and introspection.

7. **Domestic Space with Ambiguity**: Create a domestic setting that carries a sense of ambiguity and mystery. The scene could suggest multiple interpretations, inviting viewers to engage with the painting's deeper meanings, much like the Victorian fascination with The Arnolfini Portrait's ambiguous narrative.

By weaving these elements into a new painting, you can create a piece that honors the historical connection to The Arnolfini Portrait and the Pre-Raphaelites while offering fresh perspectives and engaging viewers in a rich visual dialogue.

**Example 2**

**Long Context**

This past year BART has been working its way through the environmental review process for the planned extension to Livermore. The goals of this process were to select a preferred alignment alternative from among the many considered and to preserve necessary right-of-way. A draft Program Environmental Impact Report was released last fall, which provided preliminary discussion about a slew of potential alternatives for extending BART east of its Dublin/Pleasanton terminus. An additional alignment, Alternative 2B, emerged from the public comment that was provided on the draft document. This spring, the Livermore Planning Commission, Livermore City Council, and Tri-Valley Regional Rail Policy Working Group unanimously endorsed Alignment 2B, which is a hybrid of other alternatives included in the document. In particular, it combines the two station sites from Alternative 2A with Alternative 3's Portola Avenue subway. This morning, the BART Board of Directors unanimously supported Alternative 2B as the preferred alignment and certified the the Program EIR. Although BART to Livermore has been discussed as a possibility for decades, today's vote puts the project closer than ever to being a reality. Still, a great deal of difficult work remains, not the least of which is funding. BART to Livermore, Alternative 2B. Courtesy of BART.
......

**Instruction**

Please help me create a FAQ section in English for my blog that focuses on the BART extension to Livermore. The content should be SEO-optimized, feature primary keywords in each answer, and provide clear, concise explanations suitable for readers with a basic understanding of public transportation projects.

**Response**

**FAQ: BART Extension to Livermore**
**Q1: What is the BART extension to Livermore project?**
A1: The BART extension to Livermore is a planned project to extend the Bay Area Rapid Transit (BART) system from its current terminus at Dublin/Pleasanton to Livermore. This extension aims to improve connectivity and support regional transit-oriented development.
**Q2: What is the preferred alignment for the BART extension to Livermore?**
A2: The preferred alignment for the BART extension to Livermore is Alternative 2B. This route combines features from other proposed alternatives, including a downtown subway station and an extension to the Lawrence Livermore National Laboratory.
**Q3: Why was Alternative 2B chosen for the BART extension?**
A3: Alternative 2B was chosen because it offers the best potential for long-term value by promoting transit-oriented development, providing easy access to key locations like downtown Livermore and Lawrence Livermore National Laboratory, and facilitating intermodal transfers with ACE trains.
**Q4: What are the estimated costs for the BART extension to Livermore?**
A4: The estimated cost for the BART extension to Livermore under Alternative 2B is approximately $3.83 billion. This includes expenses for new track, stations, and a maintenance yard and shop.
**Q5: How will the BART extension impact Livermore residents?**
A5: The BART extension is expected to enhance public transportation options, reduce vehicle miles traveled, and support regional development goals. However, there may be concerns about noise, construction impacts, and property acquisitions, which will be addressed in the project-level environmental review.
**Q6: What are the benefits of extending BART to Livermore?**
A6: Benefits of extending BART to Livermore include improved connectivity to the Bay Area, support for transit-oriented development, reduced greenhouse gas emissions, and enhanced access to key employment centers like Lawrence Livermore National Laboratory.
**Q7: What challenges does the BART extension to Livermore face?**
A7: Challenges include securing the necessary funding, addressing community concerns, mitigating environmental impacts, and coordinating with other regional transit projects like ACE and high-speed rail.
**Q8: What is the timeline for the BART extension to Livermore?**
A8: The timeline for the BART extension to Livermore includes multiple steps, such as completing the project-level environmental analysis, securing funding, and finalizing design and construction plans. It is a multi-year process with many milestones to achieve before completion.

**Example 3**

**Long Context**

**Document 1:**
Perhaps the most famous 16th-century portrayal of doña Marina, this description is also the most extensive from the period. Díaz del Castillo claims she was beautiful and intelligent, she could speak Nahuatl and Maya. Without doña Marina, he says, the Spaniards could not have understood the language of Mexico. These words, while evocative, were written decades after Díaz del Castillo marched with Cortés on Tenochtitlan, and thus represent both his memory of doña Marina and his reply to accounts of the conquest written and published by others. Source: Díaz del Castillo, Bernal. Chap. 22-23 in The Discovery and Conquest of Mexico, 1517-1521. 1585.
......

**Document 2:**
Tag: Visit Seville The majestic Seville Cathedral in the sultry Andalusian capital is an astounding sight that awes visitors by its sheer size and glorious splendour. The sprawling interior presents immensity, grandeur and beauty, with towering and massive column arches, ribbed vaults along with various multicoloured stained-glass windows throughout the cathedral. A collection of art treasures and liturgical items provides a glimpse of the opulence and richness the city of Seville enjoyed from the expeditions during the New World. The largest in Christendom of Gothic style in the world and the third largest in Europe following the Neoclassical Saint Peter's Basilica in Vatican City and St Paul's Cathedral in London, Seville Cathedral is an awe-inspiring architectural marvel. Built on the site of a former 12th century mosque, the magnificent Cathedral is home to a mind-blowing 30-metre tall altarpiece gilded with the finest gold brought back during the New World discovery era. The interior is adorned with masterpieces of Spanish painting and precious religious objects.
......

**Instruction**

Compare the roles and significance of Doña Marina in the Spanish conquest of Mexico and the Seville Cathedral in the history of Spain, highlighting their contributions and historical contexts.

**Response**

Doña Marina, also known as La Malinche, played a crucial role in the Spanish conquest of Mexico primarily as an interpreter and intermediary. She was highly valued for her linguistic skills, being fluent in both Nahuatl and Maya, which allowed her to communicate effectively between the Spanish conquerors led by Hernán Cortés and the indigenous populations. Her contributions were instrumental in facilitating the Spanish understanding of local customs, languages, and political landscapes, which significantly aided their conquest efforts. Beyond her linguistic abilities, her intelligence and quick adaptation to new situations made her a key figure in the Spanish campaigns. Historical accounts, particularly from Bernal Díaz del Castillo, emphasize her importance and describe her as a woman of great presence and influence among both the Spanish and the indigenous peoples.

In contrast, the Seville Cathedral, known formally as the Cathedral of Saint Mary of the See, represents a monumental achievement in Gothic architecture and a symbol of Spain's religious and cultural history. Built on the site of a former Almohad mosque, the Cathedral's construction began in 1401 and was completed over a century later in 1506. It stands as a testament to the wealth and grandeur of Seville during the period following the Reconquista, when Christian forces reclaimed the city from Muslim rule. The Cathedral is not only an architectural marvel but also a repository of significant art, religious artifacts, and the final resting place of notable figures such as Christopher Columbus. Its towering structure, intricate designs, and historical significance make it a key landmark in Spain's cultural heritage and a UNESCO World Heritage Site.

While Doña Marina's contributions were more directly related to the success of the Spanish conquest and the immediate interactions between cultures, the Seville Cathedral symbolizes the broader historical and cultural shifts in Spain, reflecting the transition from Muslim to Christian dominance and the flourishing of Gothic architecture and Christian art. Both figures and structures encapsulate pivotal moments in Spanish history, serving as enduring symbols of their respective eras and the complex interplay of cultures during those times.

## I PROMPTS

The prompts used by the WildLong framework can be seen from Table 11, Table 12, Table 13, and Table 14.

## J LIMITATION

This work aims to advance long-context instruction tuning by introducing a scalable framework for generating diverse, realistic synthetic datasets, along with comprehensive evaluations across challenging benchmarks. While our focus is on improving long-context reasoning capabilities, we acknowledge two areas where further refinement is possible: enhancing content safety and strengthening robustness against jailbreaking attacks.

To further enhance safety, future work may incorporate additional post-processing techniques such as toxicity classifiers (OpenAI, 2024; Google Jigsaw, 2024), heuristic-based filters heuristic-based filters (Albalak et al., 2024), or reinforcement learning from human feedback (RLHF) (Dai et al., 2024). Additionally, addressing vulnerabilities to jailbreaking attacks where adversarial prompts can bypass model safeguards could involve implementing strategies like prompt filtering, adversarial training, and red teaming approaches (Peng et al., 2024a; Zhang et al., 2023).

Future research may also consider broader assessments that incorporate ethical, safety, and societal alignment dimensions to complement the technical contributions presented here.

## K LICENSES

Our framework relies on two publicly available datasets—SlimPajama and WildChat—to support instruction generation and long-context pairing. We respect the licenses and terms of use associated with each dataset, and we detail them below for transparency and reproducibility.

SlimPajama is a deduplicated and filtered corpus derived from online sources. It is released under the Apache 2.0 License, which permits reuse with proper attribution and compliance with the license terms. More information is available at: `https://huggingface.co/datasets/cerebras/SlimPajama-627B`.

WildChat is a collection of user-chatbot conversations intended for training and evaluating conversational models. It is released under the ODC-BY License, which requires attribution when using or sharing the data. Details are available at: `https://huggingface.co/datasets/lmsys/wildchat`.

## L BROADER IMPACT

Improving long-context reasoning in language models can benefit applications requiring deep understanding of extended documents, such as legal analysis, academic research, and public policy review. Our scalable framework enables more accessible and diverse instruction tuning, potentially advancing the general utility of language models in complex, information-dense settings.

At the same time, enhanced long-context capabilities may increase the risk of misuse, such as generating more coherent disinformation or contextually rich manipulative content. Additionally, the generation process-though filtered using Azure OpenAI's content moderation system—may still reflect biases inherited from the underlying LLMs and source corpora. We emphasize the importance of future research on transparency, robustness, and responsible deployment practices to mitigate such risks.

Below is a conversation between a user and an AI Language Model, likely involving a long document.

**Conversation**
{conversation}

**Your Tasks**
Based on the conversation above, try to finish the following tasks.
- Determine whether the query of the user involves a long document (or any form of long text).
- If the conversation involves a long document, analyse the conversation and provide the following information using concise phrases.
  - Document Type: Specify the format or category of the document, such as a research paper, technical report, fictional story, instruction manual, etc. Ideally, extract one document type. However, if you believe there are multiple types, limit the number to two.
  - Tasks or Requests: Identify 1 to 3 the specific tasks the user wants the chatbot to perform given the long context. This may include summarizing key points, integrating multiple pieces of information, continuing the dialogue or story, providing an analysis, or any other specific task relevant to the long text. - Purpose of Query: Define the objective behind the user's query, such as educational purposes, decision-making, research, entertainment, etc. List 1 to 3 items.
  - User Intention: Determine the underlying goal or reason behind the user's request, such as completing an assignment, preparing for a debate, gaining a general understanding, etc. List 1 to 3 items.
  - User Profile: Describe the possible characteristics and background of the user in 1 to 5 phrases.
  - User's Language Style: Identify the language style of the user. List 1 to 3 items.
  - Context: Describe the situational background influencing the query, such as working on a group project, preparing for an exam, etc. List 1 to 3 context items.
  - Knowledge/Commonsense Involved for User: Identify the prior knowledge or commonsense the user is expected to have. List 1 to 5 items.
  - Knowledge/Commonsense Involved for Chatbot: Identify the prior knowledge or commonsense the chatbot is expected to have to address the query. List 1 to 5 items.
  - Long Context Capability Involved: Determine the comprehension and information processing skills required to address the user's request, such as long document comprehension, key information retrieval, handling multiple perspectives, etc. List 1 to 3 items.
  - Output Format: Identify the desired format of the response. List 1 to 3 items.
  - Sentiment: Determine the expected emotional tone or attitude in the response. List 1 to 3 items.
  - Constraint of the Request: Identify the limitations or additional requirements that the user has for the chatbot's response. List 0 to 3 constraints, if any.
  - Simplified Instruction by User: Provide a simplified version of the user's request, removing any context or background information.

**Output Format**
Document Type:
1. doc type 1 ...
2. doc type 2 ...

Task or Request:
1. request type 1 ...
2. request type 2 ...
...

**Additional Requirements for Output**
- Analyze the entire conversation to produce your answers, taking into account both the user's and the chatbot's contributions. Do not limit your analysis to just one side.
- If the user query does not involve a long document (or any form of long text), output only "No long document involved".
- For each output field, output commonly used phrases or short sentences in academic or industry if applicable.
- If you cannot extract anything for a particular field, output "NA" for that field.

Table 11: The prompt to extract meta information with GPT-4.

You are tasked with generating 3 realistic user queries or instructions for a chatbot about a long document. The user is interacting with a long {doc_type}, but you do not have access to the exact content of the document. Your task is to create reasonable user queries or instructions that meet specific meta information criteria. There are 12 meta information categories that define the characteristics of a user query or instruction. You will be provided with 6 key meta information fields that must be incorporated into each of your generated queries or instructions. For the remaining 6 categories, you have the flexibility to explore different possibilities to create varied and diverse queries or instructions. You will be given an example meta information criteria and a corresponding sample query or instruction to help you understand the context and how to apply the meta information.

**Additional requirements**
- Incorporate All Key Fields: Aim to integrate all 6 key meta information fields into each query or instruction you create. If a field is particularly challenging to include, substitute it with a reasonable alternative.
- Ensure Coherence and Creativity: Your generated queries or instructions should be coherent, natural, and flow smoothly. They should not appear as a direct combination of the meta information fields, instead aiming for a realistic scenario that a user in the given context might actually encounter.
- Creative Interpretation: The meta information criteria represent high-level characteristics of a user's query or instruction. You can interpret and apply them creatively to generate a range of realistic and diverse outputs.
- Output Format: Present your generated queries or instructions in bullet points, formatted as follows:
1. query 1
2. query 2
3. query 3

**Definitions of the 12 meta information categories**
- Tasks or Requests: tasks the user wants the chatbot to perform given the long context.
- Purpose of Query: the objective behind the user's query, such as educational purposes, decision-making, research, entertainment, etc.
- User Intention: the underlying goal or reason behind the user's request, such as completing an assignment, preparing for a debate, gaining a general understanding, etc.
- User Profile: the possible characteristics and background of the user.
- User's Language Style: the language style of the user.
- Context: the situational background influencing the query, such as working on a group project, preparing for an exam, etc.
- Knowledge/Commonsense Involved for User: the prior knowledge or commonsense the user is expected to have.
- Knowledge/Commonsense Involved for Chatbot: the prior knowledge or commonsense the chatbot is expected to have to address the query.
- Long Context Capability Involved: the comprehension and information processing skills required to address the user's request, such as long document comprehension, key information retrieval, handling multiple perspectives, etc.
- Output Format: the desired format of the response.
- Sentiment: the expected emotional tone or attitude in the response.
- Constraint of the Request: the limitations or additional requirements that the user has for the chatbot's response.

**Example meta information criteria**
{example_meta_info}

**Example query/instruction**
{example_instruction}

**Your task**
Generate a new query or instruction that aligns with the given meta information criteria:
{path_meta_info}

Table 12: The prompt to generate instruction given a sampled meta-information path.

**Long Document:**
{long_doc}

**Example Query/Instruction:**
{example_instruct}

**Your Task:**
You have been provided with a long document above, along with an example query or instruction that was formulated for another similar long document.
Your task is to create a new query or instruction that can be addressed using the information contained within the long document provided.
The new query or instruction should be inspired by the structure and intent of the given example but is not a direct copy. You should adapt the query or instruction to fit the context of the long document while still addressing a similar type of task.
Once you have formulated the query or instruction, provide a response based on the content of the long document.

Please format your output as follows:
Query/Instruction: {{query_or_instruction}}
Response: {{response}}

Table 13: The prompt to generate instruction-response pairs.

The following are tasks or requests made by users when querying a chatbot about a single document. Modify the tasks or requests as if the user is querying multiple documents. Ensure that the modifications reflect a realistic need to handle information across multiple sources, incorporating cognitive operations usually applied to multiple documents.
The document type is {doc_type}. Avoid simply adding phrases like "across multiple documents." Instead, adapt each task to reflect a more complex interaction with multiple sources, focusing on the cognitive operation that makes sense in the multi-document context.

**Cognitive operations**
- Comparison: identifying similarities, differences, or evaluating multiple documents
- Synthesis: integrating information from multiple sources to create a new, cohesive understanding
- Aggregation: collecting and presenting information from multiple sources without integrating or interpreting
- Verification and Validation: cross-referencing and fact-checking across documents
- Consensus Analysis: identifying agreement across documents
- Divergence Analysis: recognizing conflicting or differing points of view
- Problem Solving: formulating solutions based on multiple documents
- Decision Making: formulating decisions based on multiple documents
- Exploration: discovery across multiple sources without a predefined goal
- Trend and Pattern Identification: detecting larger patterns or trends from multiple documents
- Hypothesis Generation: forming new hypotheses through integrated data
- Creative Synthesis: fostering novel ideas or concepts from the documents

**Original tasks or requests** {original_tasks_or_requests}

**Output format**
1. {original_tasks_or_requests}: {modified_tasks_or_requests}
2. {original_tasks_or_requests}: {modified_tasks_or_requests}
...

Table 14: The prompt to convert single-document tasks to multi-document tasks.

