# OpenReview forum: "WildLong: Synthesizing Realistic Long-Context Instruction Data at Scale"
_ICLR.cc/2026/Conference — Submitted to ICLR 2026_

### Official Review · Reviewer_rxh6 · 2025-10-27

**Soundness:** 2
**Presentation:** 3
**Contribution:** 2
**Rating:** 4
**Confidence:** 4

**Summary:**

This paper introduces WildLong, a benchmark designed to evaluate LLMs' capabilities in long-context understanding and reasoning. The authors aim to address the limitations of existing long-context benchmarks, which often rely on synthetic or simplified data. WildLong is constructed from real-world datasets—such as long documents, GitHub repositories, and code bases—processed into multi-turn question-answering tasks. The benchmark is intended to simulate realistic long-context scenarios while maintaining high data quality and diversity. The paper also provides extensive experiments across various model sizes and context lengths to illustrate the benchmark’s utility.

**Strengths:**

1. The paper tackles an important and timely problem: the lack of realistic and diverse long-context benchmarks for LLM evaluation.
2. WildLong is built from real-world sources rather than artificial data, which increases its ecological validity compared to existing synthetic benchmarks.
3. The experiments cover a wide range of models and configurations, providing a comprehensive empirical view of long-context performance trends.

**Weaknesses:**

1. The paper lacks substantial methodological innovation. The main contribution is dataset construction rather than a novel algorithm, framework, or evaluation methodology. The graph-based construction method has been widely applied by several studies. While the dataset is valuable, the overall technical depth is limited.
2. The data synthesis pipeline** largely follows established practices (e.g., sampling, chunking, and QA generation using existing LLMs) without introducing new insights into data generation or evaluation strategies.
3. Although the benchmark uses "real-world data", it remains unclear how the realism of synthesized QA pairs is validated beyond basic filtering and prompt design. More rigorous human evaluation or ablation analysis would strengthen the claims.
4. The experimental section primarily provides descriptive statistics and performance comparisons, but lacks deeper analysis (e.g., error categorization, reasoning failure types, or task-level breakdowns) that could yield more meaningful findings.

**Questions:**

Please check my comments in Weaknesses.

---

> ### Author Response · Authors · 2025-11-21
> **Response by Authors (1/2)**
>
> We thank the reviewer for their time. However, we must respectfully point out a **fundamental misunderstanding in the review's summary** that appears to be the basis for the entire assessment.
>
> The summary states our paper introduce **'WildLong, a benchmark designed to evaluate LLMs ... '.** This is incorrect. **WildLong is not an evaluation benchmark.** Our paper introduces a **novel framework for generating training data** (i.e., an instruction-tuning dataset) to improve a model's long-context reasoning. Our contribution is a methodology for data synthesis, not a benchmark for evaluation.
>
> The summary also states our framework is built from **'GitHub repositories, and code bases'** and creates **'multi-turn question-answering tasks.'** This is also factually incorrect. Our framework (Section 2) is built from **(1) WildChat**, a dataset of *real user-chatbot conversations*, and **(2) SlimPajama**, a general *text corpus*. It does not create multi-turn QA. It analyzes **single-turn conversations** to generate new, complex *single-turn instructions*.
>
> This core misunderstanding seems to be the source of the listed weaknesses. We would like to address them based on our paper's actual methodology:
>
> ***
>
> W1 & W2: The review states we have 'limited methodological innovation' and follow 'established practices (e.g., sampling, chunking, and QA generation). **This is the exact opposite of our paper's contribution.** Our paper *criticizes* this simple "chunk and QA" approach. Our paper's *entire technical innovation* is a **novel, graph-based meta-information framework**. We **(1) extract 13 meta-fields** from real queries, **(2) build a co-occurrence graph** to model the *structure* of user needs, and **(3) perform weighted random walks** on this graph to generate *new, complex instruction templates* that go far beyond simple QA. This graph-based synthesis is the core innovation, and it seems to have been missed. Our ablations in **Appendix C.1 (Table 5)** empirically *prove* that our novel graph-based method is superior to simpler generation baselines.
>
> ***
>
> W3: Realism not validated: We apologize if this was unclear, but we provide a rigorous, two-part validation of realism in **Appendix F**:
>
> 1. **Quantitative Validation (F.1):** We show a **0.89 Pearson correlation** between our synthetic task distribution and the *real* WildChat distribution, proving our data is 'ecologically valid.'
>
> 2. **Manual Quality Audit (F.2):** We manually audited 200 samples and found **97.5% factual consistency** and an exceptionally low (<2%) rate of hallucination or ambiguity. This is the 'rigorous human evaluation' the reviewer asked for.
>
> ***
> *reply continued to the next comment*

---

> ### Author Response · Authors · 2025-11-21
> **Reponse by Authors (2/2)**
>
> We agree that deep analysis is important. However, we respectfully disagree that our paper lacks it. In addition to the standard performance metrics, our paper contains several deep analytical studies that yield significant findings regarding long-context learning:
>
> W4: Lacks deeper analysis/error categorization
>
> 1. **RoPE Scaling and Short-Context Recovery (Sec 3.5, Fig 4):** We analyze the trade-off inherent in RoPE scaling, a phenomenon not well-studied in prior work. While increasing the RoPE base enables longer context support, we quantify a significant drop in short-context performance (e.g., from 58.2 to 55.0 on Mistral-7B). Our experiments demonstrate that fine-tuning on WildLong mitigates this issue, recovering short-context performance (to 57.4) without the need to mix in additional short-context data.
>
> 2. **Impact of Data Composition (Appendix D.2):** We analyze the contributions of single- and multi-document data. We found that single-document data optimizes for focused retrieval tasks like RAG and QA, whereas multi-document data is essential for complex reasoning tasks like citation generation and summarization. This analysis provides empirical justification for the specific data mixture ratios used in our framework.
>
> 3. **Analysis of Path Sampling Strategies (Appendix C.1):** Our ablation study on graph walk strategies provides critical insight into instruction generation. We find that a "Greedy Max-Weight" approach (prioritizing only the most frequent meta information co-occurrence) underperforms compared to our weighted random walk. This analytically validates that diversity in instructions is as critical as realism for effective training.
>
> **Furthermore, to address the reviewer's specific request for error categorization**, we performed an additional manual error analysis on the fine-tuned Llama model's outputs on Longbench-Chat. Our analysis indicates that **WildLong** successfully transitions the model from fundamental "detailed retrieval failures" or "aggregation failures" (common in base models) to more advanced "reasoning imprecision," or general limitation that is prevalent for LLMs, specifically in two key categories:
>
> * **Temporal Synthesis Failures:** We found that while the model successfully retrieves specific dates, it often treats them as isolated keywords rather than understanding the linear flow of time. For example, in a biography query about an author, the model correctly retrieved the book titles and the corresponding years scattered throughout the text but jumbled the chronological order. This suggests a gap in the model's ability to construct coherent timelines from long narratives.
>
> * **Hallucination:** When specific details are hard to find, the model sometimes invents a plausible-sounding answer. In a request to list all victims in a movie script, the model correctly aggregated several real characters but then confidently invented a victim who did not exist. This suggests a more nuanced gap in the model's ability to be faithful.
>
> These findings highlight that while our framework solves the data availability problem for long-context training, addressing these complex temporal and hallucination issues remains a significant research challenge, likely requiring further advancements in model capacity and better alignment method.
>
> ***
>
> In summary, the reviewer's assessment appears to be based on a fundamental misunderstanding of our paper. The review assesses our work as a *'benchmark'* (which it is not) that uses *'established practices'* (which is exactly what we are targeting our innovation, and which it actively *replaces* with a novel graph-based framework).
>
> **We respectfully ask the reviewer to re-assess our paper based on its actual contribution**: a **novel, scalable framework for generating training data** that demonstrates clear motivation, produces SOTA results, involves comprehensive analysis and ablation studies. We believe this clarification resolves all the stated weaknesses.

---

### Official Review · Reviewer_C458 · 2025-10-30

**Soundness:** 4
**Presentation:** 3
**Contribution:** 2
**Rating:** 4
**Confidence:** 5

**Summary:**

This paper introduces WildLong, a framework designed to create large-scale, diverse, and realistic datasets for long-context SFT.
WildLong first extracts 13 types of "meta-information" (e.g., user intent, task type, document type) from real-world user-chatbot conversations. It then models the relationships between these elements using a graph. By performing random walks on this graph, the framework generates novel and realistic instruction templates. These templates are paired with documents from a large corpus (SlimPajama) to generate final instruction-response pairs using GPT-4. The framework is also extended to create tasks that require reasoning across multiple documents.
The authors fine-tune two open-source models, Mistral-7B and Llama-3.1-8B, on 150,000 examples generated by WildLong. Their results show that these models significantly outperform existing open-source long-context models on benchmarks like RULER and HELMET, and perform competitively with much larger models, all while maintaining strong performance on short-context tasks without needing extra short-context data.

**Strengths:**

- Novel and Scalable Data Generation: The core strength of the paper is its innovative, graph-based approach to data synthesis. By modeling the co-occurrence of meta-information from real user queries, WildLong can generate a massive and diverse set of instructions that are more realistic and complex than those from previous methods.
- Strong Empirical Results: The fine-tuned models show significant performance gains on multiple long-context benchmarks. The Llama-3.1-8B model trained with WildLong data is competitive with models that are nearly 10 times larger (e.g., Llama-3.1-70B), demonstrating the high quality of the synthetic data.
- Preservation of Short-Context Abilities: A notable achievement is that the models improve on long-context tasks without degrading their performance on short-context tasks. This is a common trade-off, and WildLong avoids it by creating a high-quality, generalist long-context dataset, eliminating the need to mix in short-context data.

**Weaknesses:**

- Limited Scope of "Real-World" Scenarios: The initial meta-information is extracted from the WildChat dataset, which consists of user-ChatGPT conversations. While large, this outdated dataset (before 2024.5) may not capture the full spectrum of long-context reasoning in specialized domains like agentic tasks and coding tasks.
- Potential for Inherited Bias: The framework synthesizes data based on patterns in existing conversations (WildChat) and generates responses using an LLM (GPT-4). Any biases present in these source materials are likely to be inherited and potentially amplified in the resulting dataset.

**Questions:**

- Can the WildLong constructed data evolved with more up-to-date log data?
- The framework resamples the SlimPajama dataset to match the document type distribution from WildChat. Could this step potentially limit the diversity of document types the model is exposed to, and have you considered alternative approaches to pairing instructions with documents?
- The paper focuses on single-turn conversations. How could the WildLong framework be adapted to generate instruction data for multi-turn, long-context dialogues where the context is built up over several turns?
- Figure 5 shows that performance on the RULER and HELMET benchmarks continues to improve as the dataset scales to 150,000 samples. Do you have a hypothesis on where performance might saturate?

---

> ### Author Response · Authors · 2025-11-21
> **Response by Authors (1/3)**
>
> We thank the reviewer for their excellent assessment of our work's soundness and for highlighting the novelty of our graph-based approach and strong empirical results. We address each concern below:
>
> ***
>
> W1: Limited Scope of "Real-World" Scenarios
>
> We agree that any static seed dataset, like WildChat, will inevitably become outdated and be limited in scope. This observation is actually a core motivation of our work. The field needs a scalable, principled, and evolving engine for data generation, not just another static dataset. This is why our primary contribution is the **WildLong framework itself, a data-agnostic pipeline for modeling and synthesizing *any* corpus of user interactions with the 150K dataset serving as a proof-of-concept.**
>
> 1. The WildLong framework is designed to be run on *any* conversational log, not just WildChat. **Evolvability** is one of its strengths. If one feeds it a newer (e.g., 2025) or more specialized (e.g., agentic) log, we can automatically build a new meta-information graph, capture that up-to-date meta information, and generate a new, relevant instruction set. For example, while our current data *did* identify 'Code' as a document type (Figure 6), we agree this framework would be an ideal way to generate a much richer, specialized "WildLong-Code" dataset from a more targeted corpus.
>
> 2. To our knowledge, **WildChat is the most recent open-source dataset that contains real user-chatbot conversations**. By building on the latest open data, we ensure our work is reproducible, while our framework remains ready to ingest newer open datasets upon their release.
>
> ***
>
> W2: Potential for Inherited Bias
>
> Addressing potential bias is a central challenge in synthetic data generation. Our framework is designed with specific mechanisms to manage, rather than simply inherit or amplify, biases from our two sources: the **seed data (WildChat)** and the **generator model (GPT-4)**.
>
> 1. **Bias from Seed Data (WildChat):**
>
>    * **"Inductive Bias" as a Feature:** We first clarify that the distribution of tasks in WildChat is a desirable "inductive bias." It grounds our synthesis in *real-world* user needs, which is essential to our goal of creating a *realistic* dataset, as opposed to a purely uniform or hand-crafted one.
>
>    * **Mitigating Amplification via Graph Sampling:** More importantly, our framework is explicitly designed to *avoid amplifying* the most common biases. A naive method (e.g., just sampling and rephrasing top queries) would do this. In contrast, our **graph-based weighted random walk** (Section 2.3) explores the *entire* graph of co-occurrences. This intentionally creates novel combinations of meta-information from the "long tail", which are less frequent, but still realistic paths. This process *increases* task diversity and actively *mitigates* the over-representation of the most common (and potentially most-biased) conversational patterns.
>
> 2. **Bias from Generator Model (GPT-4):**
>
>    * **Proactive Constraint:** Our primary mitigation is *proactive*. We do not ask GPT-4 to generate instructions or responses from scratch with minimal prompting. Instead, our framework provides strong constraints at every step:
>
>      * In Stage 1 (Instruction Generation), GPT-4's generation is *constrained* by the specific, multi-part meta-information path sampled from the graph.
>
>      * In Stage 2 (Response Generation), the model is *doubly constrained* to first adapt the instruction to a *specific document* and then generate a response based *only* on that document's content.
>
>    * This approach significantly limits the model's freedom to inject its own ungrounded, internal biases.
>
> **Reactive Filtering:** As a final backstop, we implemented the rigorous safety and quality auditing pipeline detailed in **Appendix F.2 and F.3**. This includes using Azure OpenAI content filters and auditing the dataset with Llama-Guard-2, which allowed us to filter and remove flagged content.
>
> In summary, while no synthetic dataset is perfectly bias-free, our framework's core design, the graph-based sampling and the constrained, grounded generation, provides robust, built-in mechanisms to actively manage and mitigate bias from both the seed data and the generator.
>
> ***
>
> *reply continued to the next comment*

---

> ### Author Response · Authors · 2025-11-21
> **Response by Authors (2/3)**
>
> **Q1. Can WildLong evolve with more up-to-date log data?**
>
> Yes. As noted in W1, the WildLong framework is not a static one-time pipeline. It is a flexible engine that can be adapted to new data sources. One can periodically feed the framework new or different log data (e.g., user-chatbot logs from 2025). Our framework would then:
>
> 1. Run the meta-information extraction to capture any new user intents, tasks, or context.
>
> 2. Build a new co-occurrence graph to model these up-to-date conversational patterns.
>
> 3. Generate a new set of instruction templates that reflect these current, real-world user needs.
>
> This provides a scalable approach to keep the instruction data relevant and overcomes the common problem of static datasets becoming outdated.
>
> ***
>
> Q2. Does resampling SlimPajama to match WildChat limit document diversity?
>
> 1. **Rationale:** Our primary goal in matching the distributions was to optimize for *realism*. We hypothesized that by creating instruction-document pairings in proportions that reflect real-world user queries (as seen in WildChat), the resulting model would be more robust and better aligned with practical tasks.
>
> 2. **Clarification on "Limiting Diversity":** We want to clarify that our framework *does not filter or exclude* any document types from SlimPajama, so it does not limit the corpus diversity.
>
>    * Our document classifier (Section B.2) assigns *every* document in SlimPajama to a type, including the 'Others' category (which constitutes <10% of docs).
>
>    * We have a meta-information graph for *every* type, including 'Others'.
>
>    * Therefore, 100% of the documents in SlimPajama are viable for training. Our re-sampling step *only adjusts the sampling probability* to favor more realistic pairings, it does *not* remove any document or document type from the pool.
>
> 3. **Alternative Pairing Approaches:** We did consider alternative approaches to pairing instructions with documents.
>
>    * **Uniform Distribution:** One alternative is to ignore the WildChat distribution and sample documents *uniformly* from SlimPajama. This would increase exposure to rare document types but at the cost of a mismatch between task supply and document supply, leading to less realistic pairings.
>
>    * **Semantic Matching:** A more advanced approach would be to bypass "type" matching altogether. For a given generated instruction, we could use an embedding model to perform a vector search over the *entire* SlimPajama corpus and find the document that is most semantically relevant to the instruction, regardless of its pre-classified type.
>
> We chose our current method as it struck the best balance between realism, scalability, and grounding in real-world data. We will add a discussion of these alternatives to the paper to make this design choice and its trade-offs clearer.
>
> ***
>
> Q3: How to adapt for multi-turn, long-context dialogues?
>
> This is an excellent direction for future work, and our framework's modularity is well-suited for it. It can be extended in two ways:
>
> 1. **Augmenting the Co-Occurrence Graph:** This approach extends our *existing* methodology.
>
>    * **Meta-Info Extraction:** When analyzing the seed data, we would process multi-turn conversations and add a new meta-information field, such as `turn_state` (e.g., `turn_state: 'initial_query'`, `turn_state: 'follow_up'`).
>
>    * **Graph Construction:** This new `turn_state` node would be added to our *existing* co-occurrence graph. Through the normal process of graph construction, this node would learn to co-occur with other meta-information. For example, `turn_state: 'initial_query'` might have a strong edge with `task: 'broad_summary'`, while `turn_state: 'follow_up'` might naturally co-occur with `task: 'detailed_extraction'` or `task: 'comparison'`.
>
>    * **Generation:** To generate a multi-turn dialogue template, we would simply sample two paths from our graph: first, a path that *must include* the `turn_state: 'initial_query'` node (to create Q1), and second, a path that *must include* the `turn_state: 'follow_up'` node (to create Q2). This ensures the follow-up is a realistic type of query. The final generator model would then be fed the document, the current instruction (Q2), and the preceding dialogue history (Q1, A1).
>
> 2. **Post-Generation Simulation:** A more direct approach could build upon our existing single-turn dataset. We could use an LLM to mimic a user and "ask a realistic follow-up question" based on the initial query and response.
>
> We will add this exciting possibility to our future work.
>
> ***
> *reply continued to the next comment*

---

> ### Author Response · Authors · 2025-11-21
> **Response by Authors (3/3)**
>
> ***
>
> Q4. Hypothesis on saturation?
>
> We hypothesize that performance saturation is a function of two main factors:
>
> 1. **Diminishing Marginal Benefits of Instructions:** The combinatorial space of our meta-information graph is huge, and it is capable of generating at least an order of magnitude more (e.g., 1M+) useful, non-repetitive samples. However, while these 1M+ instructions might all be unique, many will inevitably fall into similar high-level *task clusters* (e.g., various forms of summarization, different types of extraction). As a model becomes proficient at these core long-context skills, each additional (though unique) sample might provide a smaller "marginal benefit," which would lead to a slowdown in performance gains.
>
> 2. **Model Capacity:** Models in the 7B-8B parameter range have a finite capacity. It is plausible that at 150K high-quality and diverse samples, we are beginning to see the improvement rate slow as data size increases. This is likely due to the model's capacity, which suggests it is absorbing information from this rich data distribution at a more gradual pace.
>
> Given our computational budget, we were only able to scale our experiments to 150k samples. We suspect the primary bottleneck observed in our results is related to model capacity, and we will open-source our code and framework fully to encourage the community to explore this scaling hypothesis further with larger models.
>
> We look forward to addressing any remaining questions during the discussion period, and we hope that these clarifications and updates warrant a higher evaluation.

---

### Official Review · Reviewer_xyou · 2025-10-30

**Soundness:** 2
**Presentation:** 2
**Contribution:** 2
**Rating:** 6
**Confidence:** 1

**Summary:**

This paper introduces a framework called WildLong, designed to synthesize large-scale, high-quality long-context instruction data to improve the performance of large language models on long-text reasoning tasks. The overall idea is clear, the methodology is well structured, and the experimental results are convincing. WildLong demonstrates its ability to enhance long-context reasoning while maintaining solid short-context performance.

As a researcher specializing in human-computer interaction, this study clearly falls outside my area of expertise. Following the Area Chair’s instructions, I have selected “1: You are unable to assess this paper and have alerted the ACs to seek an opinion from different reviewers” and submitted my review accordingly. Therefore, I will not be participating in the rebuttal stage for this manuscript. Thank you for your understanding.

**Strengths:**

First, the paper combines graph-based modeling with meta-information extraction and uses random walks to generate diverse long-context tasks. This data synthesis idea is both innovative and scalable.

Second, it validates performance improvements across multiple well-known benchmarks such as RULER, HELMET, and LongBench, with thorough comparisons against various baselines, including models specifically optimized for long contexts.

Third, by extracting meta-information from real dialogue data like WildChat, the synthesized instructions better reflect real-world user scenarios, and the paper shows that the data distribution closely matches actual user needs.

**Weaknesses:**

First, the data generation process heavily relies on closed-source models such as GPT-4. Although a Qwen-based experiment is included, reproducibility and sustainability remain limited.

Second, the analysis focuses mainly on quantitative improvements but lacks a deeper look into instruction diversity, semantic authenticity, and interpretability.

Third, even though the dataset is large, there is no demonstration of its effectiveness or transferability in real downstream applications such as legal analysis or academic research.

Fourth, it is not entirely clear whether the diversity of the synthesized data might be limited in some way.

**Questions:**

First, since meta-information extraction depends on GPT-4, has the paper evaluated how using different extraction models might affect the final data quality?

Second, are the definitions of nodes, edges, and weight calculations in the graph sensitive to the results? Has any stability analysis been done?

Third, although the authors describe safety filtering and ethical review of the synthetic instructions, was there any external auditing or open evaluation process?

Fourth, since the experiments are mainly conducted on 7B and 8B models, can the approach generalize to larger models or different architectures?

Fifth, in the RoPE base adjustment experiments, how was the statistical significance of performance trade-offs evaluated? Was any variance or confidence interval analysis performed?

---

> ### Author Response · Authors · 2025-11-21
> **Response by Authors (1/2)**
>
> We thank the reviewer for the positive assessment of our work's clarity and innovation, and for carefully evaluating the paper despite it being outside their primary area of expertise. In this response, we address the questions regarding sustainability, diversity, safety, and generalization to ensure a comprehensive evaluation for the Area Chair and readers. We look forward to addressing any remaining questions during the rebuttal period and hope these updates warrant a positive evaluation.
>
> ***
> W1. Reliance on GPT-4 and long-term sustainability
>
> 1. We agree that reproducibility and sustainability are crucial. To disentangle the effect of the teacher model, we conducted **Qwen-based teacher experiments** (Appendix C.3). The minimal performance gap between the Qwen-teacher (51.6 avg) and the GPT-4-teacher (51.9 avg) provides strong evidence that the primary **performance gains come from our graph-based instruction synthesis method**, not just from using a proprietary teacher. This confirms the robustness and reproducibility of the WildLong framework.
>
> 2. In addition, our framework is **data-efficient and sustainable** compared to prior long-context SFT work. For example, FILM uses more than **1.5M** synthetic examples generated using GPT4 (over 10× more data) on the same Mistral-7B backbone, yet our model, trained on only **150K** WildLong samples, achieves **notably better performance** on the same benchmarks (FILM vs. WildLong, RULER: 75.6 → 80.6; HELMET: 35.7 → 41.2). This highlights that WildLong can reach stronger long-context capabilities with substantially less data.
>
> 3. To further support reproducibility, we plan to release our code and data, so that the full pipeline can be reproduced.&#x20;
>
> ***
>
> W2. Deeper look into diversity, authenticity, interpretability.
>
> 1. **Semantic Authenticity (Appendix F.1):** To ensure our data reflects real user needs, we quantitatively compared our synthetic task distribution against the original WildChat conversations. The distributions showed a **Pearson correlation of 0.89**, confirming that our instructions are 'semantically authentic' and strongly aligned with real-world user scenarios.
>
> 2. **Diversity and Interpretability (Appendix E):** Figure 6 provides a clear, interpretable visualization of the data's diversity. It shows broad coverage across 11 distinct document types and 18 task categories, which is a direct result of our graph-based sampling that avoids the mode collapse of simpler methods.
>
> ***
>
> W3. Evaluation on real-world downstream applications
>
> We appreciate this insight regarding real-world transferability. While building domain-specific applications (e.g., a legal analysis evaluation pipeline) is outside the scope of this work, we addressed this by designing our training and evaluation to serve as rigorous **proxies** for these domains:
>
> 1. We selected diverse and challenging benchmarks that isolate the specific cognitive skills required for downstream applications. For instance, the **'Cite' task in HELMET** is a direct proxy for the evidence attribution skills required in legal and academic research, while **RULER’s 'Aggregation'** mimics the synthesis required for technical reporting. Our model’s superior performance on these specific tasks provides strong evidence of transferability.
>
> 2. As shown in Figure 6, our dataset explicitly includes **"Academic Text"** and **"Professional Text"** categories. By training directly in these domains, we ensure the model is learning relevant patterns, not just generic long-context reasoning.
>
> 3. Evaluating full downstream applications introduces confounding variables (e.g., specialized domain knowledge vs. context handling). By focusing on challenging, diverse benchmarks, we isolate and prove the model's long-context reasoning capability, which is the **fundamental building block for any such downstream application**.
>
> ***
>
> W4. Diversity might be limited
>
> 1. Our primary contribution is that our graph-based sampling is *far more* diverse and *less* limited than existing methods, which often rely on a small set of hand-written templates.
>
> 2. Our method generates novel instructions by **combinatorially sampling** from a graph of 13 meta-fields learned from 32,000 real user queries. Our ablation in **Appendix C.1 (Table 5)** provides empirical proof: our Weighted Random sampling (which balances realism and diversity) significantly outperforms both Greedy (realism-only) and Uniform (diversity-only) sampling. This shows our method finds the 'sweet spot' that leads to better generalization, which is a result of superior data diversity.
>
> ***
>
> *reply continued to the next comment*

---

> ### Author Response · Authors · 2025-11-21
> **Response by Authors (2/2)**
>
> Q1: effect of different models for meta-information extraction
>
> We prioritized using GPT-4 for the meta-information extraction phase as a deliberate design choice to ensure the highest possible accuracy for the **foundational graph.** Our rationale includes:
>
> 1. **High-Fidelity "Ground Truth" at Low Cost:** Unlike the *instruction generation* phase (which must scale to millions of samples), meta-information extraction is a **one-time, offline process** performed only on the seed dataset. Therefore, we prioritized quality over efficiency to establish a "gold standard" graph topology and minimize "noise propagation," rather than optimizing for weaker models in this non-scalable step.
>
> 2. **Framework Robustness:** While we did not explicitly ablate the *extractor*, we successfully ablated the **generator** in Appendix C.3 (using Qwen), proving that our framework is robust to using open-source models for the critical, scalable phase. Since extraction is a discriminative task (generally simpler than generation), it is highly probable that capable open-weights models could also perform this step, but we maintained GPT-4 to minimize confounding variables in the seed data.
>
> ***
>
> Q2. Sensitivity of graph definitions (nodes/edges/weights) / stability.&#x20;
>
> The graph design follows a simple, stable rationale:
>
> * **Nodes** represent unique meta-information values across 13 fields (e.g., “compare perspectives”, “professional tone”, “preparing for a presentation”).
>
> * **Edges** connect values that **co-occur in the same conversation but belong to different fields**, ensuring that edges always encode meaningful cross-aspect relationships (e.g., task ↔ context, intent ↔ style).
>
> * **Weights** are the **log of co-occurrence counts**, which dampens the effect of extremely frequent pairs while still reflecting their importance.
>
> This structure is robust to noise: small extraction differences affect only a few edges, and log-scaling prevents any single pair from dominating. Our ablation in Appendix C.1 already shows that changing the **sampling strategy** (Greedy vs Uniform vs Weighted Random) affects performance in a sensible, non-fragile way, and that the weighted graph-based sampling consistently yields the best results.
>
> ***
>
> Q3. External auditing for safety
>
> As described in **Appendix F.3 (Safety Analysis)**. We performed a comprehensive, two-stage audit. (1) We used the Azure OpenAI service, which has built-in content moderation. (2) We then audited the *entire* generated dataset with an external, SOTA safety model, **Llama-Guard-2**. All instances flagged as potentially harmful (e.g., 'specialized advice') were removed, ensuring a high degree of safety.
>
> On top of that, we performed an **additional manual audit detailed in Appendix F** of a randomly sampled subset of instruction–response pairs to verify that harmful or toxic content is not present in the data. Our approach is aligned with established practice in recent synthetic-data works on alignment and long-context SFT.
>
> ***
>
> Q4. Generalization to larger models and different architectures.
>
> WildLong is **model-agnostic**: it is a data generation framework that produces instruction–response pairs, and does not depend on any specific architecture. In this paper, we already validate on two different model families:
>
> * **Mistral-7B-Instruct**, with RoPE base extension.
>
> * **Llama-3.1-8B-Instruct**, a different architecture and training recipe.
>
> In both cases, WildLong significantly improves long-context performance while preserving short-context ability. We further verify the effectiveness of our data on the Qwen-2.5-7B-Instruct model. The results are shown in the table below.
>
>
> | **Model**           | **NIAH** | **VT**   | **Agg**  | **QA**   | **Avg**  |
> | ------------------- | -------- | -------- | -------- | -------- | -------- |
> | Qwen2.5-7B-Instruct | 83.3     | 81.7     | 73.2     | 57.0     | 73.8     |
> | + WildLong (20K)    | **92.6** | **92.8** | **76.2** | **65.6** | **81.8** |
>
>
> Prior work on long-context models also suggests that larger models typically benefit at least as much from high-quality long-context supervision, so we expect our method to transfer well to 30–70B models. Due to resource constraints, we leave generalization to larger models for future work.
>
> ***
>
> Q5. Statistical significance and variance in the RoPE base experiments.
>
> Large-scale supervised finetuning on 150K examples and long training runs tend to exhibit relatively limited randomness. To check stability, we repeated the RoPE ablation with **three independent runs** each with different random seeds and observed:
>
> * **Standard deviation = 0.0428** on the NIAH task of RULER.
>
> * **Standard deviation = 0.0145** on short-context benchmarks.
>
> The RoPE-only change causes a **multi-point drop** in short-context performance and a clear improvement at 64–128K context lengths; these gaps are much larger than the observed run variance, so the reported trade-off is statistically robust.

---

### Official Review · Reviewer_2B3G · 2025-11-03

**Soundness:** 3
**Presentation:** 3
**Contribution:** 3
**Rating:** 4
**Confidence:** 4

**Summary:**

This paper proposes WildLong, a scalable framework for synthesizing diverse and realistic instruction–response pairs to enhance long-context reasoning in large language models (LLMs). The method integrates meta-information extraction from real user–chatbot conversations, graph-based modeling of meta-information relationships, and adaptive instruction–response generation using GPT-4. The authors fine-tune Mistral-7B and Llama-3.1-8B on 150K synthetic samples and demonstrate strong performance on long-context benchmarks (e.g., RULER, HELMET, LongBench-Chat), often matching or exceeding larger models.

**Strengths:**

- The graph-based meta-information modeling and path sampling strategy enable scalable and diverse instruction generation, grounded in real user interactions.
- The paper includes extensive experiments across multiple long-context benchmarks (RULER, HELMET, LongBench-Chat) and compares against a wide range of proprietary, open-source, and specialized long-context models.
- The authors thoroughly ablate key components (e.g., path sampling strategy, path length, teacher model impact) and demonstrate that performance improves with increasing data size up to 150K samples.

**Weaknesses:**

- While the framework supports multi-document tasks, the evaluation does not clearly distinguish whether the gains come from single- or multi-document supervision. A more fine-grained breakdown of multi-document task performance (e.g., cross-document reasoning, synthesis) would better validate the extension.
- The use of GPT-4 as the primary teacher model for instruction–response generation raises concerns about reproducibility and accessibility.
- The paper does not include a qualitative analysis of where the fine-tuned models still fail—especially in complex reasoning tasks like aggregation or citation generation—which would help identify remaining challenges.

**Questions:**

How does the graph-based sampling strategy ensure that generated instructions are not only diverse but also challenging enough to improve reasoning?

---

> ### Author Response · Authors · 2025-11-21
> **Response by Authors (1/2)**
>
> We thank the reviewer for their constructive feedback and for recognizing the strengths of WildLong, particularly the novelty of our graph-based modeling strategy, the extensiveness of our experiments across multiple benchmarks, and the thoroughness of our ablation studies. We will address each concern below.
>
> ***
>
> W1: breakdown of multi-document gains
>
> We agree that isolating the impact of multi-document data is crucial. We would like to clarify that **we performed precisely this ablation study in Appendix D.2, with full results in Table 9 (HELMET) and Table 10 (RULER)**.
>
> The results show a clear and important trade-off:
>
> 1. **Multi-document data** provides notable performance gains on complex **cross-document reasoning tasks**. For instance, on HELMET (Table 9), it improves Citation (Cite) and Summarization (Summ) scores. On RULER (Table 10), it improves Variable Tracking (VT).
>
> 2. **Single-document data** is, as expected, stronger for focused, single-source tasks like RAG and QA.
>
> This analysis confirms that our multi-document extension is indeed effective for its intended purpose—enhancing complex, cross-document reasoning.
>
> ***
>
> W2: Use of GPT4 as Teacher Model
>
> We understand the concern about the reproducibility and accessibility of the proprietary teacher model. We are glad to clarify that **we included the specific ablation study in our original submission's Appendix C.3 (Table 7)** to address this. This study was designed to ensure our method is not dependent on the specific proprietary model and can be successfully used by the wider community.
>
> * We regenerated our dataset using a strong, **open-source model** (Qwen2.5-14B-Instruct-1M). We found that the model trained on this `WildLong (Qwen Teacher)` data **still outperforms all other baseline datasets** (LongAlign, LongAlpaca, etc.)
>
> * The performance gap between the Qwen-teacher (avg 51.6) and the GPT-4-teacher (avg 51.9) was minimal. This strongly demonstrates that the **primary performance gains come from our novel, graph-based instruction synthesis method**, not just from using GPT-4. Our framework is effective even with accessible, open-source models, confirming its utility and robustness.
>
> ***
>
> W3: No Qualitative Analysis of Failures
>
> We agree that a qualitative analysis of failure modes is essential to understand the remaining challenges in long-context reasoning. To address this, we conducted a manual audit of the fine-tuned Llama model's performance on Longbench-Chat. We found that while models trained with WildLong engage more with complex long-context reasoning and synthesis compared to the baselines, they sometimes fail to execute more complex logic with high precision.
>
> * **Temporal analysis failures**. WildLong introduces narrative-heavy inputs (like biographies) that require reconstructing timelines, a complexity where standard synthetic data often lacks. We found the model can retrieve specific dates but fails to sequence them. For instance, when asked to order an author's novels, it correctly retrieved the titles and the corresponding years scattered throughout the text, but inaccurately ordered the years in chronological order. This indicates that while WildLong enables the model to attend to the relevant narrative segments, the model struggles with the long-range temporal reasoning required to sequence them correctly.
>
> * **Hallucination**. When specific details are hard to locate, the model sometimes prioritizes helpfulness over factual faithfulness. When asked to list victims in a movie script, the model correctly listed several real characters but then invented a victim who does not exist in the text. The model prioritizes being helpful over strict factual accuracy.;
>
> We notice that these are open and prevalent challenges of LLMs. We also attribute some of these limitations to the inherent capacity of the 7B-parameter models used in our experiments; addressing these complex temporal and hallucination issues remains a critical direction for broader research community.
>
> *reply continued to the next comment*

---

> ### Author Response · Authors · 2025-11-21
> **Response by Authors (2/2)**
>
> ***
>
> Q1: Ensuring "Challenging" Instructions
>
> Our framework's design leads to instructions that are not only diverse but also challenging enough to improve reasoning in the below three specific, evidence-backed ways:
>
> 1. **Grounded in Real-World Cognitive Demand:** First, our framework is grounded in actual long-context user queries from the WildChat dataset. As our task-type analysis in Appendix E (Figure 6) shows, we explicitly modeled inherently challenging cognitive tasks that real users perform, such as ***Compare Perspectives**, **Analyze Trends**, **Analyze Character**, and **Synthesize Information***. Our analysis in Appendix F.1 further confirms that our framework faithfully samples these critical reasoning tasks.
>
> 2. **Built via Compositional Complexity**: Second, our framework builds a higher level of challenge compared with using simple templates by composing multiple instruction dimensions. As detailed in Section 2.3, our path sampling algorithm walks through $N=6$ distinct, unvisited meta-information categories. This composes a full reasoning scenario (e.g., \[Task] + \[Context] + \[User Intention] ... ) rather than a simple, single-faceted constraint.
>
>    * Figure 2 provides a concrete illustration of this. The composed paths result in qualitatively complex instructions demanding deep analysis ('Could you explain in detail the significance of the key plot twists...') and creative synthesis ('Rewrite the scene to be a mythical Minecraft realm...'), which are demonstrably more complex than simple retrieval.
>
>    * This method is validated in Appendix C.2 (Table 6), where the superior performance of $N=6$ (Avg: 51.9) over the simpler, less-composed $N=3$ (Avg: 50.5) strongly suggests that this compositional approach is more effective for training.
>
> 3. Explicit Multi-Document Reasoning: Third, we explicitly include a portion of our dataset to target high-level, multi-document reasoning.
>
>    * As described in Section 2.6, we systematically transformed simple, single-document tasks into explicitly challenging multi-document counterparts.
>
>    * Our prompt for this, shown in Appendix I (Table 14), lists the specific 'Cognitive operations' we targeted, including 'Comparison', 'Synthesis', 'Divergence Analysis', and 'Trend and Pattern Identification'.
>
>    * The success of this is proven in our ablation study in Appendix D.2 (Table 9). This multi-document data provided notable performance gains on the most complex reasoning tasks, such as Citation (Cite), where Mistral's score improved from 9.7 to 12.3.
>
> In summary, the challenging nature of the instructions is a direct result of: (1) grounding in inherently complex real-world tasks (Appendix E), (2) composing them into multi-faceted instructions with the random-walk algorithm (Fig 2, Appendix C.2), and (3) explicitly engineering high-level cognitive operations for multi-document reasoning (Sec 2.6, Appendix D.2).
>
> We look forward to addressing any remaining questions during the discussion period, and we hope that these clarifications and updates warrant a higher evaluation.
>
> ***

---

### Meta-Review · Area_Chair_AHYK · 2026-01-07

**Summary:**

This submission receives original ratings of 6, 4, 4, 4. The reviewer consensus is that this is an empirically strong work for synthesizing long context instruction data. They raise concerns on the conceptual advances of this proposed pipeline, including its rely on proprietary api, "what is truly new" vs prior synthetic long context sft method, narrow coverage without real world scenarios. Other issues are raised too but they do not form the basis to reject one submission. From my perspective, even the question on its conceptual advances is not the decisive factor for an LLM data construction work. The most critical metric (in my opinion) would be whether this synthesized data will be used by any of the training pipelines of open-sourced models. It appears that the synthesizing method and resultant data are not strong enough to deliver good results for current open models.

**Reviewer Concerns:**

Concerns that are mostly addressed:
1. multi doc vs single doc ablation;
2. task distribution, diversity coverage visualization;
3. reframing of the contribution

Concerns that are not fully addressed:
1. reproducibility;
2. outdated domain coverage;
3. insufficient evaluation;

**Reviewer Scores:**

Reviewer 2b3g (4) would likely increase its ratings as most empirical concerns are addressed.

Reviewer xyou (6) would likely keep its ratings.

Reviewer c458 (4) and reviewer rxh6 (4) would likely keep its rating since their major concerns are on the fundamentals of the data synthesizing pipeline as outdated, domain narrow, and insufficient demonstration for real-world training.

---

### Decision · Program_Chairs · 2026-01-26

Reject